# Rolling contact fatigue calculation of a three-row roller pitch bearing in a wind turbine

Oliver Menck[1], Florian Schleich[1], and Matthias Stammler[1]

[1]Fraunhofer Institute for Wind Energy Systems IWES, 21029 Hamburg, Germany

**Correspondence:** Oliver Menck (oliver.menck(at)iwes.fraunhofer.de)

**Abstract.** Rolling contact fatigue calculations of wind turbine pitch bearings have to consider the oscillating operation and the complex, dynamic load distribution in the bearing. The present work proposes a methodology to calculate the life of a pitch bearing that is a roller bearing, specifically a three-row roller bearing. Previous publications in this field have discussed the life calculation of ball pitch bearings. Methodologies applicable to any pitch rolling bearing are partly improved upon in the present work. Several aspects not applicable to roller bearings are re-thought. In comparison to ball bearings, the rolling contact fatigue calculation of roller bearings adds another level of complexity, since they have a long contact which is typically discretized if strong deformation is present, leading to significantly more degrees of freedom. Exemplary calculations are carried out using a slightly modified version of an extensively validated FE model of a three-row roller bearing of a wind turbine. This paper focuses on the methodology used to calculate the life. The axial rows were found to have a much lower fatigue life than the radial row, and thus the axial rows are the main determinant of the fatigue life of the bearing. They also were shown to have a much lower uncertainty than the radial row using the approach proposed in this paper.

## 1 Introduction

Modern wind turbines contain a pitch bearing (also called rotor blade bearing) at the root of each rotor blade which is used to rotate the blade along its longitudinal axis. This is used to start and stop the turbine and to control the loads acting on it (Burton et al., 2011). While pitch bearings in older turbines mostly performed relatively little movement, newer turbines increasingly use control mechanisms that are intended to reduce fatigue loading on various components of the turbine by dynamically adapting the pitch angle during operation (Bossanyi, 2003). This increases movement of the pitch bearing and therefore decreases its rolling contact fatigue (RCF) life.

RCF is a fatigue damage type that is caused by rolling elements in a rolling bearing repeatedly moving over the raceway. Movement of the rolling element, even without any friction, causes changes in sub surface shear stress. This shear stress, particularly the orthogonal shear stress, is thought to be the initiating stress that causes RCF (Lundberg and Palmgren, 1947), though other stresses have been proposed to be the origin of RCF as well (Sadeghi et al., 2009). Cyclic changes in sub surface stress near inclusions or material defects cause microcracks that grow into larger, macroscopic spalls. These can cause failure of a bearing. Fatigue is a stochastic phenomenon; the life of the bearing until a spall occurs can therefore only be determined

for a given probability. Industry-typical is the use of $L_{10}$, which denotes the life for 90 % reliability, i.e., the operational time until which 90 % of bearings do not have any visible spall damage on their raceways.

     Pitch bearings can experience a number of failure mechanisms (Stammler et al., 2020). Some of these are relatively untypical failure modes that specifically affect pitch bearings due to their unusual operational behavior, such as ring fracture, which occurs due to the variation in external bending moment loading (Becker et al., 2022), or wear damages on the raceway, which

can be caused by the oscillatory movement patterns of the bearing (Schwack, 2020; Stammler, 2020). Rolling contact fatigue on the other hand is a more common type of damage that can, in principle, affect any bearing in any operational scenario if the loads are sufficiently high and the bearing moves (rotates or oscillates) sufficiently. The calculation methods for rolling contact fatigue are thus standardized in ISO (see ISO 281, ISO/TR 1281-1, ISO/TR 1281-2, ISO 16281), although a number of other approaches also exist in the literature. Nonetheless, for the application in wind turbine pitch bearings, a number of additional

questions arise, particularly due to the flexibility of the bearing rings, which necessitate finite element (FE) simulations, and due to the stochastic variation in loads and movement due to the wind.

     Although there are a number of pitch bearing types that are used by different manufacturers, the most common type is probably the double-row four-point bearing. Publications on pitch bearings reflect this fact as the vast majority of them cover double-row four-point bearings, see for instance (Schwack et al., 2016b, a; Stammler et al., 2018; Lopez et al., 2019; Menck

et al., 2020; Leupold et al., 2021; Menck et al., 2022; Keller and Guo, 2022; Graßmann et al., 2023; Menck, 2023; Rezaei et al., 2023; Escalero et al., 2024; Rezaei et al., 2024; Schleich et al., 2024). Far fewer publications have been written on the subject of roller bearings as pitch bearings, see (Stammler et al., 2018) for a load calculation of a roller pitch bearing and (Stammler, 2023) for a wear test program on roller pitch bearings. While still less common than double-row four-point bearings, anecdotally, roller bearings seem to become more common types of pitch bearings for increasingly larger turbines.

While many of the aforementioned publications on double-row four-point bearings present or apply life calculation approaches (Schwack et al., 2016b; Lopez et al., 2019; Menck et al., 2020; Leupold et al., 2021; Keller and Guo, 2022; Menck, 2023; Rezaei et al., 2023; Escalero et al., 2024), there are almost no publications on the life calculation of roller bearings aside from the NREL DG03 (Stammler et al., 2024), which is a guideline for pitch bearing calculation, and industry presentations that give limited detail on the calculation approach (Becker, 2024). Roller bearing life calculations include an additional degree

of complexity compared to ball bearing life calculations, because rollers are profiled in order to prevent edge stresses (Harris and Kotzalas, 2007). Profiles are manufacturer-dependent and their use necessitates numerical contact models, since simple Hertzian equations cannot accurately determine the stresses that occur in a roller-raceway contact; they may only approximate it for ideal contact conditions (Lundberg, 1939).

     The given paper aims to show a comprehensive approach for the life calculation in a three-row roller type pitch bearing in

a modern wind turbine. A three-row roller bearing contains two axial rows, one for positive axial loads and one for negative ones, and a radial row for radial loads. It is depicted schematically in Fig. 1 with the raceway nomenclature used in this paper. The turbine used for the calculation is multi-MW product. Several information as bearing dimension, roller profile, calculation results cannot be shared for confidentiality reasons. The bearing model used for this paper is also slightly modified in a manner

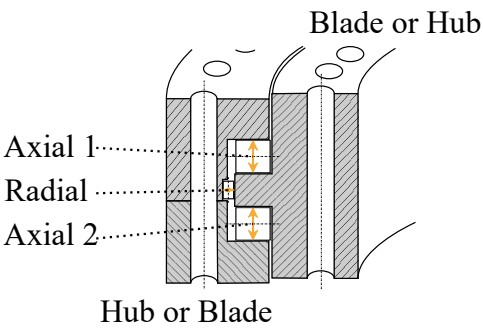

**Figure 1.** Example three-row roller bearing with raceway nomenclature and hub and blade locations. Hub either attaches to either inner or outer ring, blade attaches to the other ring.

that is confidential. Instead normalized results are shown and roller profiles are defined using ISO profiles according to ISO 16281.

The calculation approach shown herein follows closely the abovementioned NREL DG03, the rolling contact fatigue calculation approach of which is based closely on ISO/TS 16281 (now replaced by ISO 16281). Some previous papers have criticised ISO standards for supposedly being unreliable for large scale pitch bearings, see (Lopez et al., 2019; Potočnik et al., 2010), but no evidence of this has been given in the literature known to the authors. Results from previous life calculations (Schwack et al., 2016b; Menck et al., 2020) of double-row four-point bearings were so absurdly low that they could not possibly be accurate. At the time of writing, the authors assume that this discrepancy can be done away with by modifying smaller aspects of the presented life calculation method, like for instance by applying different life modification factors. There is no evidence known to the authors that the life calculation based on ISO is qualitatively unreliable for pitch bearings.

In the following, the life calculation approach for three-row roller bearings is discussed. Section 2 discusses the FE simulation model used for this paper, and the ways in which it has been validated. Using this model, a grid of FE simulations was calculated for interpolation of all load cases of the turbine. The choice of this grid is discussed in Sect. 3. Section 4 then discusses how rolling contact fatigue life for individual load cases is calculated here and calculates it for the FE grid points from the previous section. Section 5 then finally covers how these results are processed to obtain a pitch bearing life representative of the entire turbine operational time. Finally, Section 6 concludes the paper.

## 2 FE simulation

Accurate FE simulations are critical to a realistic life calculation. This section therefore covers the three-row roller bearing FE model including a short insight into the bearing validation process and the rotor FE model that was used to simulate the resulting load distribution for a variety of turbine load situations.

## 2.1 FE bearing model and validation

The three-row roller bearings implemented in the rotor model are created using the commercial ANSYS extension Rolling Bearing inside ANSYS (RBiA). The RBiA extension implements the rolling elements between the predefined raceway surfaces by COMBIN39 nonlinear spring elements that connect to the raceways by surface-based constraints. 21 springs are used to represent one single roller because this was the maximum possible amount of springs that could be used in the extension at the time of writing this paper. The force deflection curves of the nonlinear spring elements are calculated according to the defined curvatures of the contacting roller and raceway according to ISO 16281. A crowning of the axial and radial rollers is considered according to the same standard. Furthermore, the bearing model considers clearance or preload of rows by means of PRETS179 elements. Negligible geometric details like small holes or lubrication channels are removed in order to enable a smoother mesh generation. High order tetrahedral elements with quadratic shape functions and mid nodes are used for the meshes of the bearing rings. As it is typical for three row roller bearings, one of the inner or outer rings splits up into two rings as it is necessary for the assembly process. The split may occur on the inner or outer ring, depending on the bearing design. In Fig. 1, it is located on the outer ring. The location of the split for the bearing in this paper is confidential, Fig. 1 is not necessarily representative. This split of the ring is considered in the model and the surfaces are connected to each other by an internal frictional contact. The blade bearing model was validated with measured strain data obtained from a real scale test on the BEAT6.1 test bench of the Fraunhofer IWES. This test bench tests two bearings simultaneously and applies the loads with six hydraulic actuators in hexapod configuration. Adapter components emulate the bearing's surrounding stiffnesses. Figure 2 shows the BEAT6.1 with an exemplary test setup for 5 m blade bearings. More information about the test environment can be taken from Stammler (2020). Strain gauges are attached equidistantly to the blade bearing's inner and outer ring as depicted in Fig. 3.

For the validation process, static tests were performed with pure bending moments. As the strain gauge signal fluctuates over time, the data is averaged over the time the applied load of the test bench is kept constant. Furthermore, according to the recommendation in Graßmann et al. (2023) for a blade bearing validation, in some tests the applied bending moment was kept constant while the blade bearing was pitched. The measured data of the different strain gauges is plotted against the simulated data for the maximum of the ramped loads. Whereas the measured data is only available at certain circumferential positions, the FE simulated data is postprocessed along the entire circumference via path operations considering a high number of sample points. Either the axial or the tangential strains are postprocessed depending on the path's position. Figure 4 shows the comparison of axial strain values for inner and outer ring at a high $M_x$ bending moment whereas Fig. 5 shows the comparison for the tangential strain values.

Even if there are slight discrepancies at certain positions, the overall characteristics of the deformation behaviour match well between simulation and test. Figure 6 shows results for a constant $M_y$ bending moment for the outer rings of both test bearings.

The scatter bars in the plots indicate the fluctuation in the measured signals which is caused by the pitch movements of the bearing. This finding was already described in Graßmann et al. (2023) in a more pronounced way for scaled blade bearings. It can be seen that the scatter bars due to the pitch movements can be differently pronounced between both test bearings.

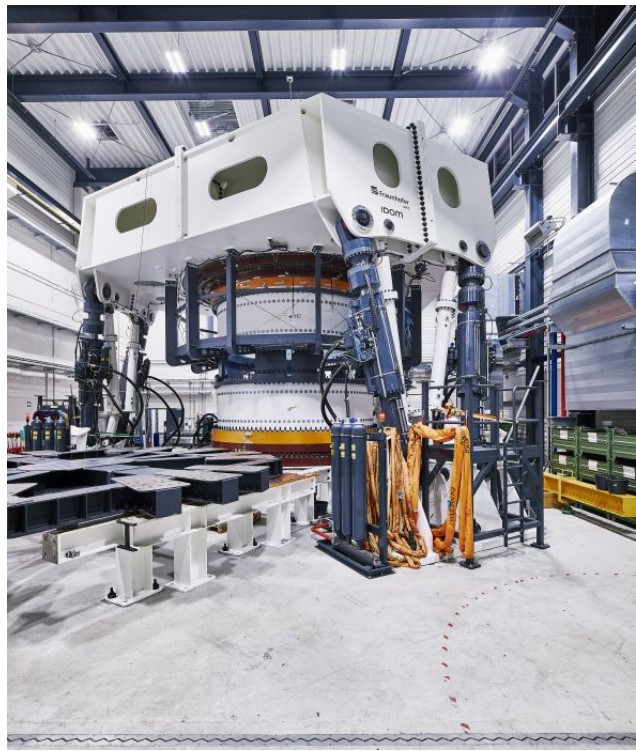

**Figure 2.** BEAT6.1 test bench ©Fraunhofer IWES/Marcus Heine

However, the deviations between simulation and test are in the same range as the scatter caused by the pitch movements. One possible explanation for the slight deviations between simulation and test could be, that only the bearing bolted connections are considered in this stage of the BEAT6.1 FE model in order to save computational time. Later on, some investigations with different test assemblies have shown that test rig internal bolted connections can also affect the bearing strain results up to this extent. However, as the focus was on the validation of the bearing model and there is a good agreement between simulation and test results for the global deformation behaviour, it is assumed that with this bearing FE model realistic raceway deformation and tilting is simulated as well, which is crucial for the subsequent bearing lifetime calculations.

## 2.2 FE rotor model

Like the bearing FE model, the global FE rotor model is built up in the ANSYS workbench environment. The focus is put on accuracy and on efficiency of the model in order to be able to perform a high number of simulations in a reasonable time. The high number of simulations is required in order to evaluate the resulting bearing load distributions for a variety of different load situations required for the advanced subsequent lifetime calculations. The rotor model contains the rotor hub, three blade dummies, one blade bearing at the primary flange (1), two blade bearing dummies at the secondary flanges (2, 3) and bearing

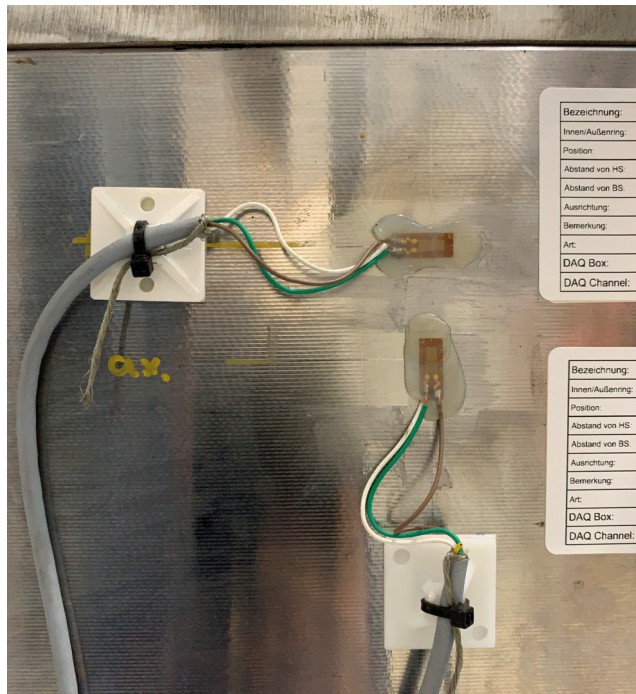

**Figure 3.** Strain gauges on the roller bearing's outer ring.

stiffening components at all three flanges. Figure 7 shows an exemplary FE rotor model which is built up in a comparable manner.

Only the blade dummies used for the present investigation are cylindrical and shorter as the blade structures depicted in Figure 7. The three blade dummies are meshed with a hex-dominant method to reduce the number of nodes and elements for these components. Bolted connections are considered only at the primary flange. They are implemented with BEAM elements that connect to washers which are modeled with 3D solid elements. The desired preload of both bolted connections is implemented with PRETS179 elements in the first load step. Only at the primary flange frictional contacts are considered between the component flange surfaces and for the contacts between component flange surface and washer surface. Different frictional coefficients are used depending on the flange surface individual coating. A general joint with the connection type "Body-Ground" is applied to the hub's rotor flange to fix the entire model and implement a certain stiffness to account for the rotor shaft. A so called "pitch lock" condition is implemented at the primary flange by means of a remote displacement. This boundary condition fixes only the nodal degrees of freedom in rotational $z$-direction and in turn prevents any rotatory displacements of the inner ring against the outer ring. As only the primary flange 1 is equipped with a full roller bearing model with spring elements no pitch lock boundary condition is necessary for the secondary flanges 2 and 3. The coordinate systems for the load application are shown in Figure 7. Forces and moments are applied to pilot nodes located in the centre of the blade dummy cutting plane at each flange. This lever arm between blade bearing flange and pilot node position is considered

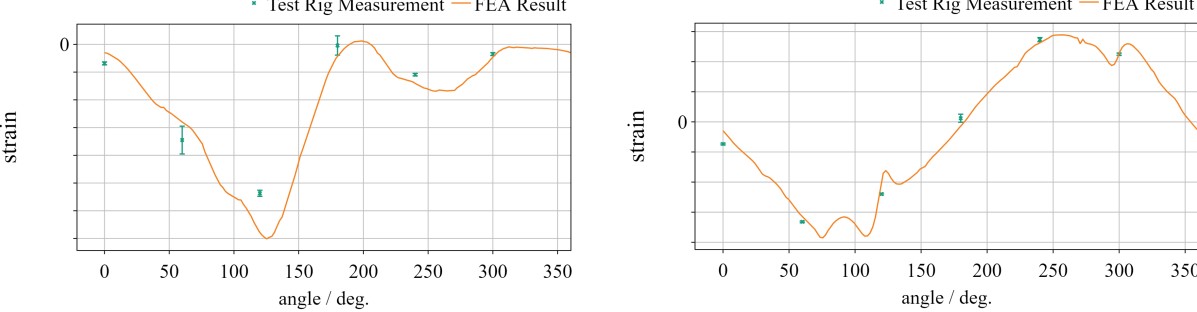

**Figure 4.** Axial strain for $M_x$ bending moment for inner (left) and outer ring (right)

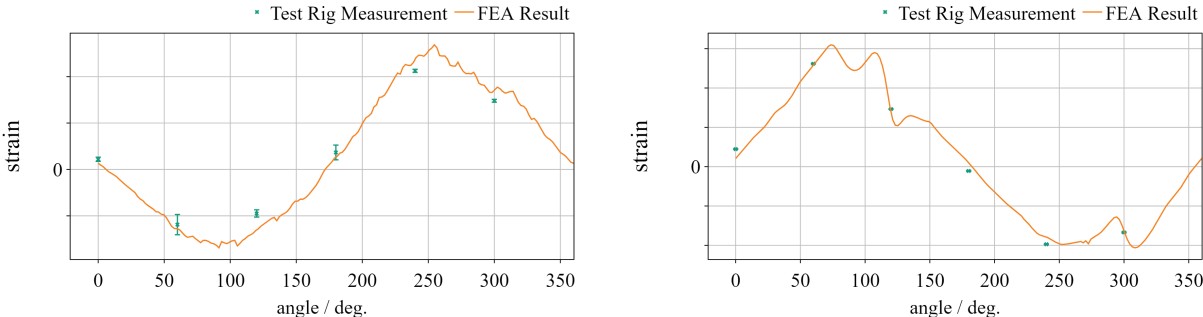

**Figure 5.** Tangential strain for $M_x$ bending moment for inner (left) and outer ring (right)

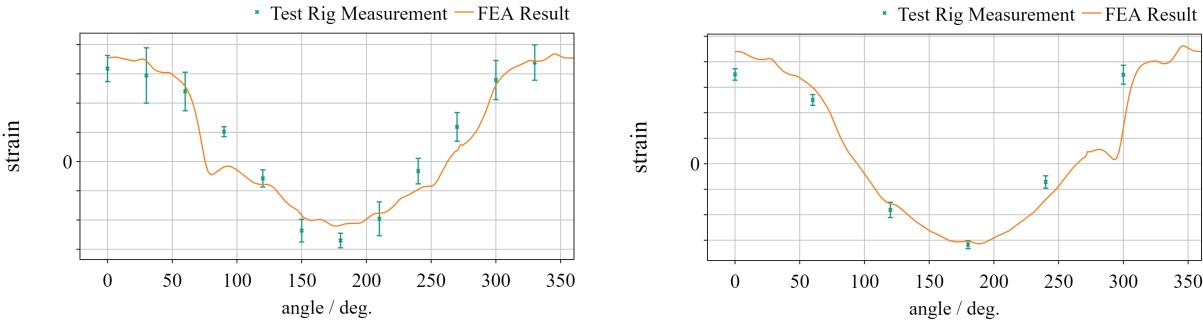

**Figure 6.** Outer ring tangential strain for $M_y$ bending moment with pitch movement. Lower test bearing (left) and upper test bearing (right).

accordingly. Probe functions were used to verify the force and moment reaction at the blade bearing flange nodes. In addition, plausibility checks of the resulting load distributions were done for a variety of different load scenarios.

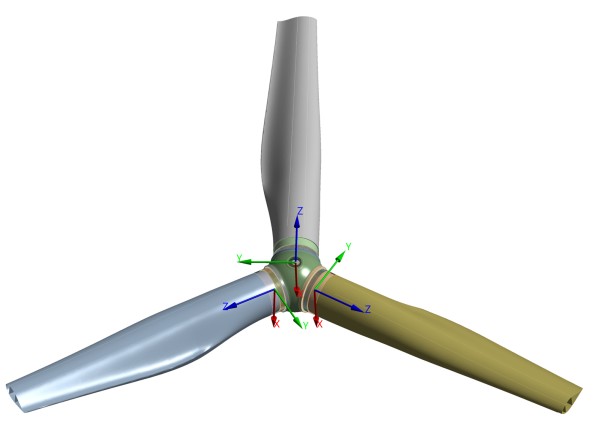

**Figure 7.** Exemplary FE rotor model of the IWT7.5 reference turbine.

## 3 FE simulation grid

Loads of the turbine have been calculated with aero-elastic simulations according to IEC 61400-1. Thus, there is a significant amount of time series simulation covering all power production (DLC 1.2), Fault (DLC 2.4), Start and Stops (DLC 3.1 and 4.1) and Parked conditions (DLC 6.4).

All time series were simulated using time steps of 50 ms. Taken together, these time series result in multiple millions of time steps. Post-processing these time steps into bins is a common approach to reduce the number of simulation points, used, for instance, in Schwack et al. (2016b); Menck et al. (2020); Keller and Guo (2022). However, using these time steps individually is more accurate than post-processing them into a smaller number of bins (Menck and Stammler, 2024). It is however practically impossible to calculate millions of FE simulations for the life calculation of a pitch bearing.

In order to be able to use individual time step data rather than using bins, a regression or interpolation akin to that shown in Menck et al. (2020) can be used. It is based on a reduced number of simulation points (hereafter referred to as "grid") that are simulated in FE. Contact forces or pressures of the bearing for aeroelastic loads in between these FE grid points are then determined via a regression or an interpolation.

There are 15 degrees of freedom (DOF) for the external loads in the FE model described in 2 (Note that the FE model obviously has far more internal DOF). They are $F_x$, $F_y$, $F_z$, $M_x$, and $M_y$ for each of the three blades, respectively. Because the above described FE simulations need hours to run, this high number of DOF needs to be reduced down to a lower amount of DOF for the FE simulation grid.

The following sections describe the choice of FE grid points that will be used for interpolation in the following chapters, starting with the DOF for blade 1, which are most relevant for the load distribution in that bearing, and then continuing with the loads of blades 2 and 3.

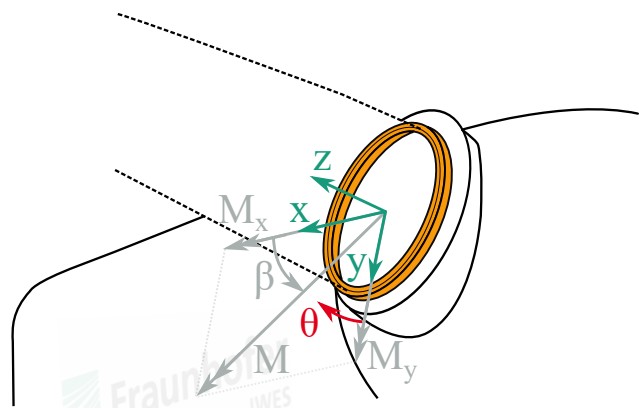

**Figure 8.** Coordinate system used for this paper, taken from Menck et al. (2020).

## 3.1 Degrees of freedom for blade 1

Pitch bearings experience significant loads in five degrees of freedom (DOF) according to the coordinate system in Fig. 8: $F_x$, $F_y$, $F_z$, $M_x$, and $M_y$. Since the sixth DOF ($M_z$) would be the rotational DOF of the bearing, it is comparatively small and can be neglected for the purposes of this paper. Aside from these DOF, the blade can also be rotated, which also influences the load distribution and can be considered as an additional DOF.

Previous publications by the authors have used the resulting bending moment $M$ (also called $M_{\mathrm{res}}$), the load angle $\beta$, and the
170 pitch angle of the blade $\theta$ in order to define a grid of FE simulation points, see Menck et al. (2020). Further in-house analyses have shown that the influence of the pitch angle is relatively small in many turbines.

For these reasons, deviating from previous publications by the authors, the pitch angle was not used as a DOF in this paper. The resulting bending moment $M$ and the load angle $\beta$ (representing $M_x$ and $M_y$) have continued to be included since they appear to be the most significant influence on the load distribution. This leaves the forces $F_x$, $F_y$ and $F_z$ to be somehow
included in the FE simulations.

The following Sect. 3.1.1 will show that $F_x$ and $F_y$ are closely correlated with $M_y$ and $M_x$ and can thus be approximated based on them. $F_z$ will be shown to correlate poorly with the bending moments and is thus included as additional DOF for the FE simulation grid. The Sect. 3.1.2 will then show that a very small number of $F_z$ variations are required for the FE simulation grid.

## 3.1.1 Approximation of $F_x$ and $F_y$ for blade 1

Forces $F_x$ and $F_y$ are the resultants of weight and aerodynamic loads distributed over the entire blade length. They are a necessary input of the FE simulations. As the theoretical center point of the resultants changes with the distribution of the acting loads, they are not ideally correlated with the bending moments $M_y$ and $M_x$ at the blade root.

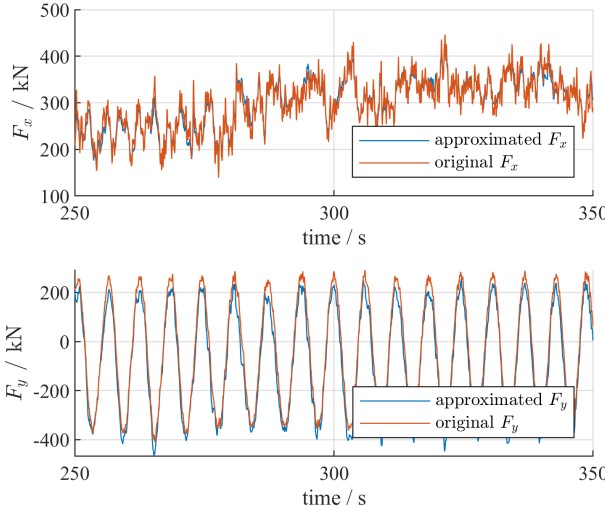

**Figure 9.** Approximations of $F_x$ and $F_y$ based on the bending moment, exemplary for IWT7.5

Despite this, there is still a close correlation between the blade root radial loads and bending moments. A linear approx-
imation will be used here to approximate the radial loads based on the bending moment components chosen later in the FE
simulation grid. To this end, all data points from the simulation were used to minimize the root mean square error (RMSE) of

$$F = a + M/b. \tag{1}$$

In order to be able to show detailed results, the authors also carried out this task using publicly available IWT7.5 aeroelastic
data (Popko). Doing so resulted in

$$F_x = 63.06\,\text{kN} + M_y/56.61\,\text{m, and} \tag{2}$$

$$F_y = 27.86\,\text{kN} - M_x/20.89\,\text{m.} \tag{3}$$

Over all simulation points of the IWT7.5 data, this resulted in an RMSE of 27.53 kN for the approximation of $F_x$ via Eq. 2
and an RMSE of 51.08 kN for the approximation of $F_y$ via Eq. 3. Figure 9 shows a sample section of a 12 m/s wind speed
time series with the real aeroelastic $F_x$ and $F_y$ being compared to their approximations based on Eqs. 2 and 3. Qualitatively,
the result can be seen to work well, and the RMSE of these methods is quite low compared to loads in operation, particularly
for the more significant load component $F_x$ which causes the flapwise bending moment $M_y$.

The same analysis was carried out using the aeroelastic data of the studied wind turbine and will be used for the FE simula-
tions shown later. Attempts to correlate $F_z$ with the bending moments ended poorly, with errors of a linear fit to the resulting
bending moment being above 300 kN for the example IWT7.5 data shown above. For this reason, the axial load $F_z$ was used
as a third degree of freedom next to $M$ and $\beta$.

### 3.1.2  Choice of grid simulation points for $F_z$ for blade 1

The axial load is often considered as a linear factor for the approximation of rolling element loads, see Stammler et al. (2024). The authors of the present work hypothesized that two FE simulations at the upper and lower end of $F_z$ might be sufficient to determine any other $F_z$ load in between these extremes.

To verify this hypothesis, ten FE simulations were carried out using the models shown in Sect. 2: Five with varying levels of $F_z$ for a relatively low bending moment in the lower 10 to 20 % of DLC 1.2 bending moments, and one in the upper 90 to 100 % thereof. The five levels of $F_z$ are normalized to -0.33, 0, 0.33, 0.67, and 1 in this paper. Upper and lower end of $F_z$ (i.e., $F_z = -0.33$ and 1) were close to actual upper and lower end seen in DLC 1.2. This resulted in load distributions for two axial rows and one radial row, all of which contained rollers that were discretized into 21 laminae for the FE simulation.

Roller lamina loads $q$ of levels $F_z = 0, 0.33$, and $0.67$ were then interpolated only using the FE load data from the extremes at $F_z = -0.33$ and 1. A linear interpolation was used. An excerpt of the result is shown in Fig. 10 for a lamina at the center of the roller and for an exemplary roller on Axial row 2. The lamina load $q$ can be seen to be represented well using the interpolation, particularly at the top of the load distribution. Merely at the lower end, discrepancies are visible; these are, however, of exponentially smaller influence to the final rolling contact fatigue life.

Qualitatively, the result appeared the same for Axial row 1 and the radial row: Higher forces were interpolated very well, only at the lower end of the load distribution, the interpolation underestimated forces. Since highly loaded rolling elements are those that affect the life the most, this result is adequate for the life calculation.

From examining these and more results of the simulations for various rollers and laminae, the authors conclude that it is valid to interpolate two extreme load levels of $F_z$ and to interpolate in between them. This works particularly well for high lamina loads, which are the most relevant for rolling contact fatigue life.

### 3.2  FE simulation grid for blade 1

Concluding from the above discussed results, three degrees of freedom are sufficient for the FE simulation grid. These include bending moment $M$, load angle $\beta$, and axial load $F_z$. The loads $F_x$ and $F_y$ are strongly correlated with $M_y$ and $M_x$ and thus do not need to be considered separately. The pitch angle has a low influence, particularly for the given study, and is therefore negligible.

The bending moment's influence on roller lamina loads $q$ is by far the biggest out of DOFs. However, it is also relatively linear in behavior. The load angle's influence is not as significant, but highly nonlinear. Therefore, a similar number of FE grid points were used for $M$ and $\beta$. Based on previous experience, the authors opted for 6 points for $M$ and 7 points for $\beta$, as well as two points for $F_z$ due to the reasons discussed above. This results in $6 \cdot 7 \cdot 2 = 84$ FE simulations that will form the basis for all calculations in this paper. For illustrative purposes, the grid has been recreated using IWT7.5 data, shown in Fig. 11.

It can be seen that no FE simulations were carried out in the range of $\beta = 0...-180°$. This is because this range represents very rare operating points which have little significance on life. The studied wind turbine data included even fewer points than the IWT7.5 shown in Fig. 11, making this choice even more appropriate. The grid shown in Fig. 11 also contains 14 FE

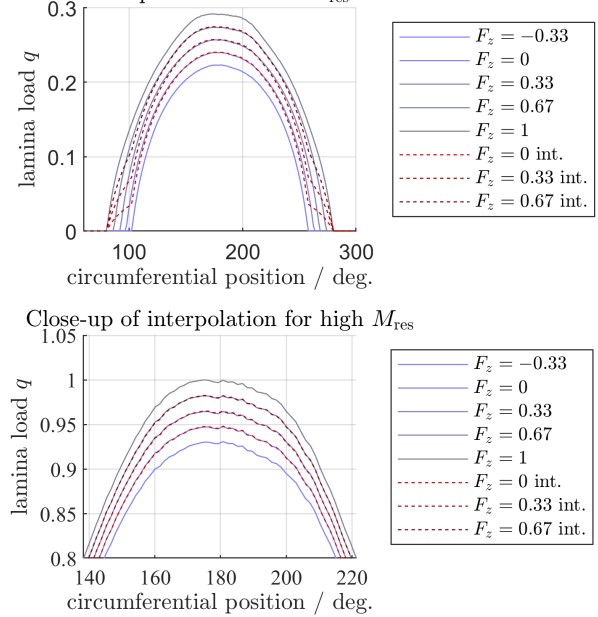

**Figure 10.** Interpolation of simulations with varying $F_z$. Center lamina load on Axial row 2. Lamina loads $q$ normalized to maximum load in the figure. $F_z$ normalized to maximum load for the present study.

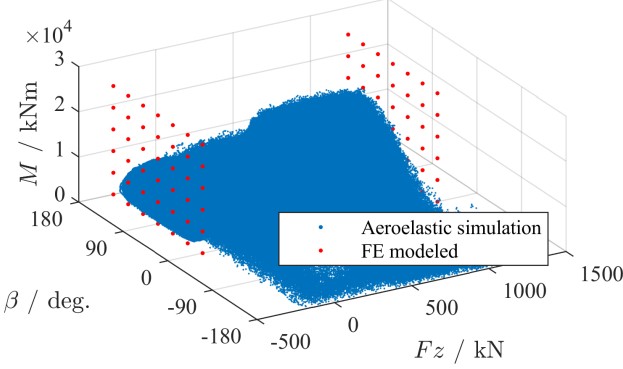

**Figure 11.** Grid of FE simulations vs. aeroelastic time steps, exemplary for IWT7.5

simulation points at a bending moment of $M = 0$, in which case the load angle is irrelevant; thus, only two of these simulations have been carried out for the high and low $F_z$ load each. Therefore, only 72 FE simulations were actually carried out. Tables A1 and A2 contain the grid data points that were simulated, where moments have been given depending on the maximum bending moment $M_{max}$ that was simulated here, and only the sign (positive or negative) is given for $F_z$.

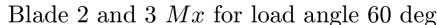

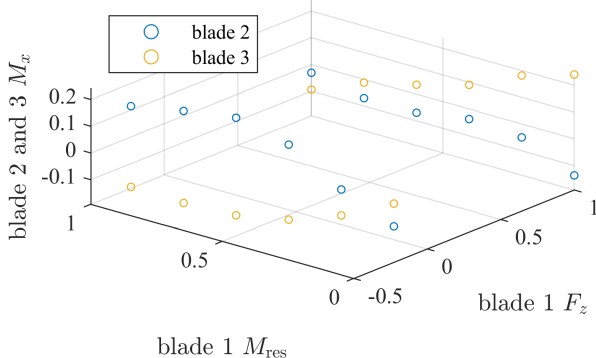

**Figure 12.** $M_x$ of blades 2 and 3 depending on $M_{res}$ and $F_z$ of blade 1 for studied turbine. Axes normalized to maximum $M_{res}$ and $F_z$.

### 3.3 Loads at blades 2 and 3

The FE model used in this paper is a full rotor model and thus also requires loads at blades 2 and 3. These influence the load distribution in the bearing, especially at circumferential positions close to other bearings, and therefore need to be determined for the simulation grid used here (Schleich et al., 2024).

In principle, this adds another 3 DOF for each of the other two blades; however, while they have a minor influence on RCF calculation results, it does not justify such a huge increase in computational expense for the scope of the present work (Note this influence of blade 2 and 3 is likely more significant for other damage modes such as ring cracks, as they influence the edgewise loads, which are known to affect ring cracks (Becker and Jorgensen, 2023)). For pragmatical purposes and because previous analyses determined that there is some influence of the other two blades, but it is less critical than that of the first blade, mean values of blades 2 and 3 were determined for the grid shown in Fig. 11. In Stammler and Schleich (2024), similar mean values are determined. While the present work uses $F_z$, Stammler and Schleich (2024) use the rotor azimuth angle $\Phi_{r,B}$. Standard deviations for the approach used in the present work have not been determined as the influences of blade 2 and 3 on blade 1 are less critical than in the case of a structural fatigue calculation as in Stammler and Schleich (2024).

An excerpt of these results is shown in Fig. 12, where some average values for $M_x$ of blades 2 and 3 are shown based on the blade 1 loads in the FE simulation grid, and in Fig. 13, where the values of $M_y$ of blades 2 and 3 are shown.

$M_y$ of blades 2 and 3 can be seen to be correlated well with blade 1, which is plausible, since all of them are loaded similarly by the wind. The correlation between $M_x$ of blades 2 and 3 with the blade 1 loads is more complex, but still plausible: At negative values of $F_z$, blade 1 points upward. This means blade 2 is generally experiencing a positive $M_x$, since it is located 120 degrees further in rotational position of the hub, where gravity is acting in its positive $M_x$ direction, and blade 3 is experiencing a negative $M_x$ since it is located yet another 120 degrees further, where gravity is acting in the negative $M_x$ direction. For cases of high postive $F_z$ values, the situation is opposite, and blade 1 pointing downwards means that blade 2 is experiencing a negative $M_x$ due to gravity whereas blade 3 a positive one.

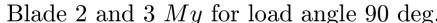

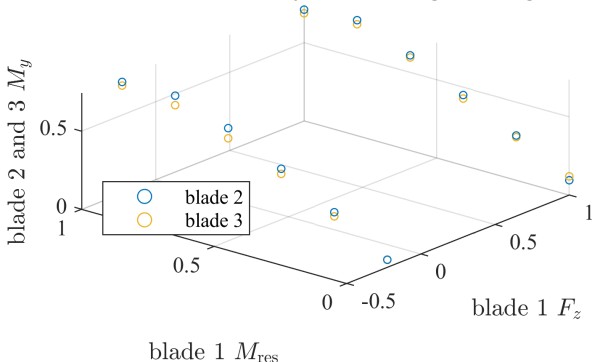

**Figure 13.** $M_y$ of blades 2 and 3 depending on $M_{\mathrm{res}}$ and $F_z$ of blade 1 for studied turbine. Axes normalized to maximum $M_{\mathrm{res}}$ and $F_z$.

## 4 Rolling contact fatigue life for individual load cases

The rolling contact fatigue life calculation adheres closely to the ISO/TS 16281 based methodology described in NREL DG03 (Stammler et al., 2024). These documents far exceed the requirements of a more simple life calculation according to ISO 281. They require a detailed pressure distribution coming from a detailed simulation of the bearing that includes rollers divided into several laminae. As described above, this aim is achieved here by usage of FE simulations. The result of this calculation is then, according to the abovementioned documents, processed with a non-Hertzian contact calculation. The non-Hertzian pressure distribution is then used to calculate the life of each raceway which are then combined to obtain a total life.

This section covers the calculation for individual load cases obtained from FE in order to carry out various analyses and checks prior to the calculation of the fatigue life of the bearing. The calculation procedure for bearing life $L_{10}$ is given in Sect. 4.1. This calculation requires contact pressures, the calculation of which is discussed in Sect. 4.2. Finally, the life of the grid of FE simulations from Sect. 3 is calculated in Sect. 4.3.

### 4.1 Calculation of $L_{10}$

The $L_{10}$ life for calculations in this section were calculated according to the NREL DG03 (Stammler et al., 2024) which closely adheres to ISO/TS 16281 (now ISO 16281). According to NREL DG03, the bearing is separated into two axial rows and a radial row.

The axial row calculation starts with the determination of their load rating $C_{\mathrm{a}}$, which is identical for both axial rows, as

$$C_{\mathrm{a}} = b_{\mathrm{m}} \, f_{\mathrm{c}} \, L_{\mathrm{we}}^{7/9} \, Z^{3/4} \, D_{\mathrm{we}}^{29/27} \tag{4}$$

where $b_m = 1$, $L_{\mathrm{we}}$ is the effective roller length (i.e., roller length minus chamfer), $Z$ is the number of rollers per axial row, and $D_{\mathrm{we}}$ is the roller diameter. The variable $f_c$ is calculated according to ISO/TR 1281-1.

For the radial roller row, the load rating $C_r$ is calculated as

$$C_{\mathrm{r}} = b_{\mathrm{m}}\, f_{\mathrm{c}} \left( L_{\mathrm{we}} \cos\alpha \right)^{7/9} Z^{3/4}\, D_{\mathrm{we}}^{29/27} \tag{5}$$

where $\alpha$ is the nominal contact angle of $0°$ for the radial row, and $b_m = 1.1$ for the radial roller row.

Using these load ratings, the variables $Q_{\mathrm{ci}}$ and $Q_{\mathrm{ce}}$ for the inner and outer ring are determined. For the axial rows, these are identical due to the $90°$ contact angle, giving

$$Q_{\mathrm{ci}} = Q_{\mathrm{ce}} = \frac{1}{\lambda\nu} \frac{C_{\mathrm{a}}}{Z} 2^{2/9} \tag{6}$$

where $\lambda\nu = 0.73$ for the axial row.

For the radial row, they are given by

$$Q_{\mathrm{ci}} = \frac{1}{\lambda\nu} \frac{C_{\mathrm{r}}}{0.378 \cdot Z \left(\cos\alpha\right) i^{7/9}} \left\{ 1 + \left[ 1.038 \left( \frac{1-\gamma}{1+\gamma} \right)^{143/108} \right]^{9/2} \right\}^{2/9} \tag{7}$$

$$Q_{\mathrm{ce}} = \frac{1}{\lambda\nu} \frac{C_{\mathrm{r}}}{0.364 \cdot Z \left(\cos\alpha\right) i^{7/9}} \left\{ 1 + \left[ 1.038 \left( \frac{1-\gamma}{1+\gamma} \right)^{143/108} \right]^{-9/2} \right\}^{2/9} \tag{8}$$

where $\lambda\nu = 0.83$ for the radial row and $\gamma = D_{\mathrm{w}} \cos\alpha / D_{\mathrm{pw}}$.

From now on, the equations are identical for axial and radial row, though results will differ due to the different dimensions of the rows and their rollers.

The basic dynamic load ratings $q_{\mathrm{ci}}$ and $q_{\mathrm{ce}}$ of a lamina for inner and outer ring are given by

$$q_{\mathrm{ci/e}} = Q_{\mathrm{ci/e}} \left( \frac{1}{n_{\mathrm{s}}} \right)^{7/9} \tag{9}$$

where $n_s$ is the number of laminae (sometimes also called slices) into which the roller is divided for the calculation. Equation 9 assumes laminae of even length. In Sect. 4.2.1, a convergence analysis is carried out in order to determine the amount of laminae used for the remainder of this paper.

Potentially occurring edge stresses are accounted for by use of a non-Hertzian model that is described in Sect. 4.2. For each roller $j$ and lamina $k$, the stress riser functions $f_{\mathrm{i}}[j,k]$ and $f_{\mathrm{e}}[j,k]$ for the inner and outer rings are given by

$$f_{\mathrm{i/e}}[j,k] = \left[ \left( \frac{p_{\mathrm{Hi/e},j,k}}{271} \right)^2 D_{\mathrm{we}} \left(1 \mp \gamma\right) \frac{L_{\mathrm{we}}}{n_{\mathrm{s}}} \right] / q_{j,k} \tag{10}$$

where $p_{\mathrm{Hi},j,k}$ and $p_{\mathrm{He},j,k}$ are the pressures on lamina $k$ of roller $j$, determined using the non-Hertzian model described in Sect. 4.2, for inner and outer ring, respectively.

The dynamic equivalent load for each lamina $k$ can now be calculated. Following NREL DG03 and Menck and Stammler (2024), the equivalent load for a stationary ring is used for both inner and outer ring, since pitch bearings oscillate by relatively small amplitudes. These small oscillations cause the oscillating ring to be almost stationary w.r.t. the load, differing from a

rotating bearing, where the rotating ring rotates relative to the load. Thus, for all raceways,

$$q_{k\text{ei/e}} = \left[ \frac{1}{Z} \sum_{j=1}^{Z} \left( f_{\text{i/e}}[j,k] \, q_{j,k} \right)^{4.5} \right]^{1/4.5} \tag{11}$$

Note that when Eq. 10 is put into Eq. 11, the lamina load $q_{j,k}$ cancels out and is therefore not actually required for the life calculation.

The life of each inner-outer raceway pair $m$, i.e. of the first axial row ($m = 1$), the second axial row ($m = 2$), and the radial row ($m = 3$), is then given by

$$L_{10\text{r},m} = \left\{ \sum_{k=1}^{n_\text{s}} \left[ \left( \frac{q_{k\text{ci}}}{q_{k\text{ei}}} \right)^{-4.5} + \left( \frac{q_{k\text{ce}}}{q_{k\text{ee}}} \right)^{-4.5} \right] \right\}^{-8/9} \tag{12}$$

Finally the life of the entire bearing is calculated using

$$L_{10} = \left( \sum_{m=1}^{3} L_{10\text{r},m}^{-9/8} \right)^{-8/9} \tag{13}$$

Up until this point, the life is measured in millions of revolutions. Consideration of the oscillatory behavior of pitch bearings will follow in Sect. 5.2.

## 4.2 Contact pressure calculations

The FE simulations are carried out using 21 laminae, resulting in 21 lamina loads $q_{\text{FE},k}$ for each roller $j$. Pressures of these laminae can be calculated using Hertzian theory. However, Hertzian theory simplifies the roller-race contact and may under-estimate the real pressure. Therefore, the resulting load and moment from these laminae are then determined for each roller $j$ according to

$$Q_j = \sum_{k}^{21} q_{\text{FE},j,k}, \tag{14}$$

$$M_j = \sum_{k}^{21} t_k \cdot q_{\text{FE},j,k}, \tag{15}$$

where $t_k$ is the distance of the center of lamina $k$ from the roller center. Since $M_j$ is the moment around the roller center, $t_k$ assumes negative values in negative direction and positive ones in positive direction. The force and moment are then used as an input to an adapted non-Hertzian contact calculation based on Reusner (1977). This returns a more accurate pressure distribution than the FE simulation and is capable of accurately detecting edge stresses that may occur in a roller and that would go unnoticed by FE simulations using the approach described in Sect. 2. Reusner proposed an approach that determines contact properties such as roller force, moment, pressure, and contact width based on the displacement and misalignment of the roller. The calculation by Reusner has been chosen because it is one of three methods that are explicitly suggested for this task by ISO 16281 and NREL DG03. It has been adapted for this paper in order to use forces and moments as input rather than as outputs, referred to as "inverted Reusner calculation" (or just "Reusner" for short) in the following.

### 4.2.1 Contact convergence analysis

For the non-Hertzian contact calculation according to Reusner (1977), the roller is evenly discretized along its length into $n_s$ laminae. The number of laminae for the FE simulation and the inverted Reusner model are unrelated and can be chosen completely independently. The amount of laminae to perform an accurate life calculation depends on the specifics of the loading that the rolling bodies are experiencing, and on the roller design, including its profile. Particularly the occurrence of edge loading or strong misalignment necessitates a higher number of laminae.

In order to determine the required amount of laminae per roller for the following life calculation, convergence analyses were carried out with the inverted Reusner calculation. To this end, the rollers were divided into $n_s$ laminae of even length. The variable $n_s$ was varied from ten to 100 in steps of ten, and then additionally once set to 150. Life of the raceways and the entire bearing were determined according to Sect. 4.1 for each of the different values of $n_s$. The roller life at $n_s = 150$ was used as a reference for the lives at lower lamina numbers.

The convergence analysis showed that the axial raceway 1 (near the blade flange) and the radial raceway both converged well. With 30 laminae, these raceways already converged within 3 % error of the final reference result at 150 laminae, with the exception of a single simulation for the first axial raceway. The second axial raceway (near the hub) converged more poorly. 8 of the 72 simulations did not yet reach less than 3 % of difference compared to the reference result at 150 laminae using 30 laminae. These are, however, mostly unrealistic load situations that are used for interpolation purposes at rare operating points only.

However, only the life of the entire bearing and its convergence really counts for the further calculations. For the life of the entire bearing and using 30 laminae, only 5 of the 72 simulations (Nr. 10, 20, 62, 71, 72) did not reach a life within 3 % of the reference result at 150 laminae. All of these five load situations represent unrealistically high cases of mostly edgewise bending moments that do not occur in reality and are only used for interpolation of rare operating points. All other simulations were within 3 % difference of the reference when using 30 laminae. Figure 14 shows results of the convergence analysis for simulation nr. 36, 38, 40, and 42. These correspond to pure flapwise bending moments with a positive $F_z$ and 40 %, 60 %, 80 %, and 100 % of $M_{max}$, respectively, see tables A1 and A2. The convergence behavior can be seen to deviate from the reference for higher lamina numbers (for instance $n = 70$ for sim. nr. 42) and then to converge towards the reference. This is due to edge stresses that occur in some contacts of the Reusner calculation for the particular calculations done in this paper with a medium amount of laminae, and it shows why a convergence analysis is necessary.

Pressure distributions for sim. nr. 42 (maximum $M_{res}$ load case, pure flapwise direction, positive $F_z$) are given in Fig. 15. All rollers of each raceway are shown and plotted on top of each other. For the axial raceways, inner and outer ring pressures are identical due to the contact angle of $\alpha = 90°$. For the radial rollers, inner ring pressures are displayed. While there is some misalignment present on all raceways, its effect is most significant on the axial raceway 2. Maximum loads all occur at the highest bending moment $M_{max}$ that has been simulated, but for different load angles thereof. As shown above in Fig. 11, $M_{max}$ represents a very rare load situation during operation, and the overall highest pressures occur at load angles that do not occur in operation for such high loads. These simulations and are only used for interpolation of rare operating points.

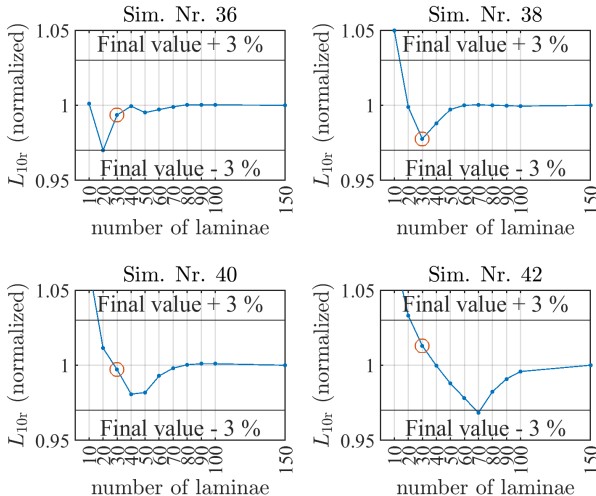

**Figure 14.** convergence analysis of $L_{10\mathrm{r}}$ for selected FE simulations (see tables A1 and A2). Result for $n_{\mathrm{s}} = 30$ laminae marked red.

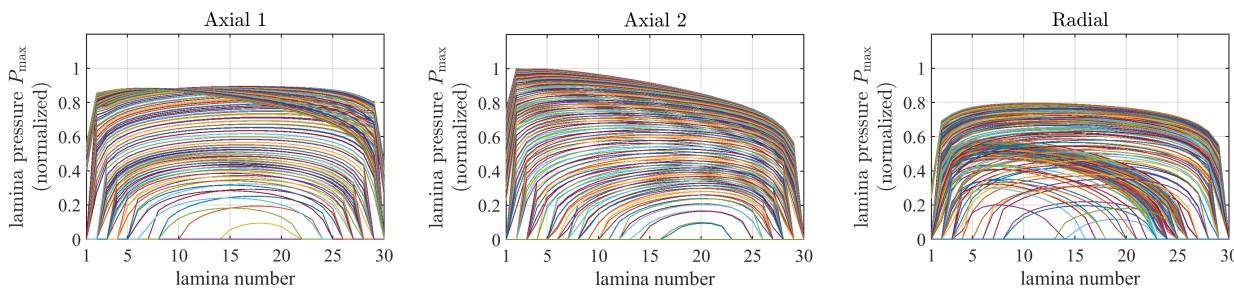

**Figure 15.** Pressure distributions of all rollers on axial raceway 1, 2, and radial raceway. Obtained via inverted Reusner algorithm. Simulation nr. 42 ($M_{\max}$, $+F_z$, and $\beta = 90$). Pressures normalized to maximum pressure within the three figures.

The pressures that have been obtained via the inverted Reusner algorithm can be compared to those directly obtained from FE using a lamina approach as described in Sect. 2. A comparison of both pressure results is given in Fig. 16. The center of the roller can be seen to have slightly higher results than the results directly obtained via FE (0.9 % higher for both axial rows and 0.5% for the radial row). Conversely, towards the edges of the roller, the FE results have much higher pressures, since the Reusner processed results decrease strongly at the outer ends (see Fig. 15) whereas the FE results barely decrease at all at the edges.

In any case, both results are very similar. This is due to the identical profiles used both in FE as well as the inverted Reusner algorithm, as well as because of the fact that no edge pressures occur. A non-Hertzian analysis for the roller pressures is particularly necessary because edge pressure spikes can occur with an unsuitable choice of profile or if there is a lot of roller misalignment. In the present case, there are no edge pressure spikes present. Therefore, the results are similar to the FE results.

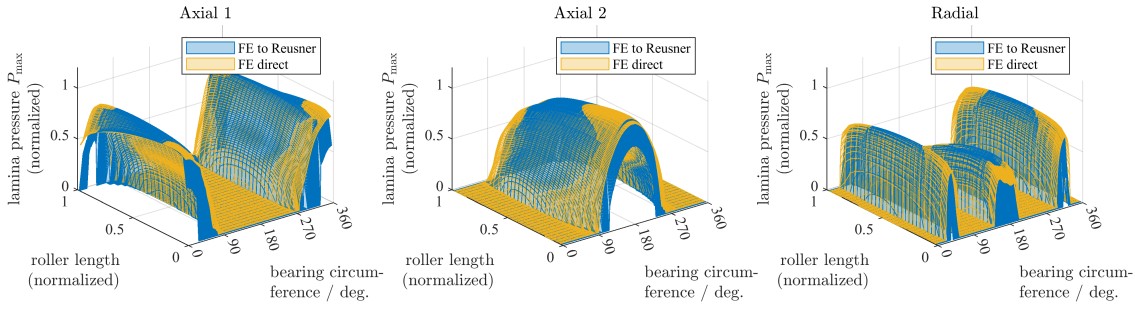

**Figure 16.** Pressure distribution of all rollers on axial raceway 1, 2, and radial raceway, directly obtained from FE as well as post-processed via inverted Reusner algorithm. Simulation nr. 42 ($M_{\max}$, $+F_z$, and $\beta = 90$). Pressures normalized to maximum pressure of Reusner processed results within the three figures.

Had the profile been less adequate, or if there had been more misalignment, there would be more deviations between the two approaches.

### 4.3 Bearing life of FE simulated cases

The lives $L_{10r}$ of all 72 simulations have been calculated according to the above equations. They are shown in Fig. 17. While all further calculations will use $L_{10r}$ with inverted Reusner post-processing, $L_{10r}$ without it, using the Hertzian pressure from FE, has also been calculated for reference. Life $L_{10r}$ has been normalized to the maximum $L_{10r}$ obtained with the inverted Reusner post-processing approach.

  The results of both approaches can be seen to be very similar, which is due to the fact that both approaches lead to similar
pressure distributions in this study, see Fig. 16 and the corresponding discussion. The lowest $L_{10r}$ lives in Fig. 17 correspond to the highest bending moments, see Tables A1 and A2. An increase in $F_z$ appears to have a positive effect on $L_{10r}$ for some simulations and a negative one on others. Further, the load angle $\beta$ appears to have a small but noticeable effect, with the simulations at $\beta = 90°$ deviating the most from the other load angles that were simulated. Because pre-tension in the bearing was considered, the rollers were loaded even at $M_{res} = 0\,\mathrm{Nm}$ (Simulation Nr. 21 and 32) and thus a finite life $L_{10r}$ is obtained
even for these simulations.

  Fig. 18 shows the lives $L_{10r}$ of all three raceways separately, normalized to the highest $L_{10r}$ among the axial rows, i.e., the life of Axial 1 in simulation 32. The radial row has the highest life $L_{10r}$ out of all three raceways, with few exceptions for some simulations. Both Axial row 1 and the radial row appear to benefit from an increase in axial force $F_z$, while $L_{10r}$ of Axial row 2 decreases with an increase in $F_z$. The whole bearing life is most affected by those raceways with a low $L_{10r}$, see Eq. 13.
Especially at the high bending moment load cases in which the total life is lowest, the whole bearing life can thus be seen to be dominated by the axial rows whose life is much smaller than that of the radial row.

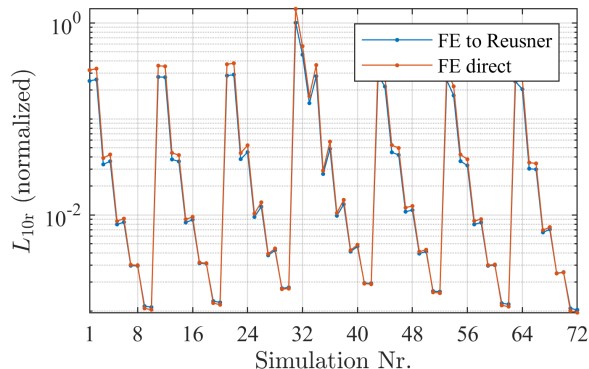

**Figure 17.** $L_{10r}$ of all 72 FE simulations, normalized. Results shown for FE results with and without inverted Reusner post-processing.

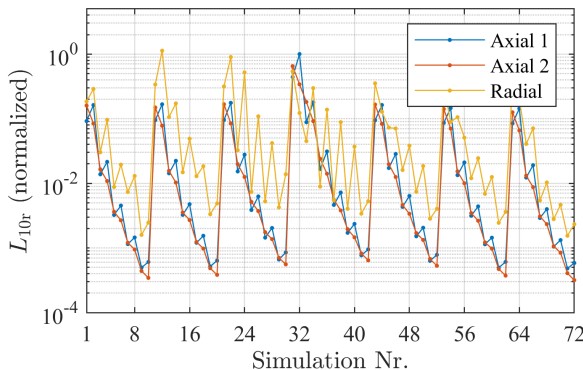

**Figure 18.** $L_{10r}$ of all raceways of all 72 FE simulations, normalized.

## 5 Rolling contact fatigue life for the entire operating time

In this section, the life of the bearing under all operating conditions that it experiences is calculated. To this end, the contact pressures in each time step of the simulation are determined. The procedure to determine contact pressures for any time step is described and tested with sample simulations in Sect. 5.1. Section 5.2 then discusses how the lives of all time steps are combined into one total operating life. Finally, Sect. 5.3 proposes a simplified method for the life calculation of a three-row roller bearing based on the previously achieved results.

### 5.1 Interpolation of pressures

The grid chosen in Sect. 3 was based on a previous publication by the authors (Menck et al., 2020) in which contact forces were determined using a regression for a double-row four-point bearing. For the present roller bearings, pressures are required

rather than forces (loads). This is because the lamina load $q_{j,k}$ cancels out when Eq. 10 is put into Eq. 11, but the pressure $p_{\mathrm{Hi/e},j,k}$ does not cancel out.

Using a regression or interpolation of forces is a likely feasible approach since contact forces are roughly proportional to the bending moment $M$ in particular, which has the highest influence on rolling element loads (Stammler et al., 2024). However, for the present roller bearing calculation, contact forces from FE must be postprocessed into an inverted Reusner algorithm, see Sect. 4.2. While the inverted Reusner algorithm is much faster than an FE simulation, it takes some seconds (approximately 30 seconds for one load case in this paper on a standard business laptop) to calculate an entire bearing load case with $n_\mathrm{s} = 30$ laminae as used in this paper, because there are hundreds of contacts in the bearing that need to be simulated.

Based on Menck and Stammler (2024), a calculation in which time steps are directly used as input is more accurate than one where post-processing of the time steps into bins takes place. Therefore, usage of time steps is the preferred choice that will be used in this paper. However, with multiple millions of time steps taken from the aeroelastic simulations for this paper (see Sect. 3), a computation with the inverted Reusner algorithm for every single step becomes too computationally expensive.

While contact force and contact pressure in a rolling bearing are not proportional, they can locally be approximated as linear. Therefore, a regression or interpolation of the pressures is a more feasible alternative to the computationally heavy and long-enduring alternative in which roller forces are interpolated (or determined via a regression) and must be processed with a non-Hertzian calculation for each time step.

Thus, instead of interpolating (or approximating via a regression) forces, pressures will be interpolated (or approximated via a regression) in the following. To this end, the grid of FE simulations is processed through the inverted Reusner algorithm to obtain a pressure distribution for $n_\mathrm{s} = 30$ laminae, based on the results in Sect. 4.2.1. The pressure in each of these 30 laminae for each roller is then interpolated (or approximated via a regression) for each time step. The approach is tested in the following using randomly selected test cases. Different approximation methods - a regression and two interpolations - are compared.

### 5.1.1 Testing the interpolation

In order to verify the interpolation approach described above, 20 test simulations were carried out. They were actual operating points from the aeroelastic simulations, randomly sampled from wind speeds between 10 and 14 m/s. The test simulations were simulated including all 15 DOF, i.e., all 5 DOF of each rotor blade, and no simplification of DOFs akin to Sect. 3 has been undertaken for these 20 test simulations.

The life of these test simulations was then calculated by processing the results into the inverted Reusner algorithm and calculating the life using the above described procedure.

Then, for verification of the interpolation, the interpolation was applied to these test simulations. Their load angle, $M$, and $F_z$ values were determined and the load distribution of these simulations was calculated. The location of the test simulations within the grid is given in Fig. 19 and Table B1. The life was then determined according to the above described procedure using the interpolated pressure distributions.

Three different approaches were used: A regression from Menck et al. (2020), a linear interpolation, and a cubic spline interpolation. The results are shown in Fig. 20. Lives are normalized to the highest life of the whole bearing using the FE

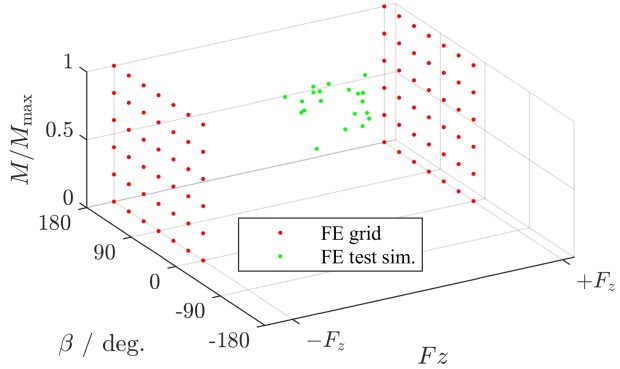

**Figure 19.** Locations of test simulations in FE simulation grid.

**Table 1.** Mean absolute percentage error of different approximation approaches compared to inverted Reusner post-processed FE results

| Method | Total bearing | Axial 1 | Axial 2 | Radial |
|---|---|---|---|---|
| regression | 18.64 % | 19.63 % | 12.88 % | 371.73 % |
| linear int. | 26.30 % | 16.28 % | 11.77 % | 235.90 % |
| spline int. | 13.71 % | 4.91 % | 2.93 % | 193.56 % |

results without interpolation or regression ("FE to Reusner"). Percentage errors (PE) were calculated for all 20 cases according to

$$\text{PE} = \frac{L_{10,\text{r,approx}} - L_{10,\text{r,FE}}}{L_{10,\text{r,FE}}} \cdot 100\% \tag{16}$$

Further, the mean absolute percentage error (MAPE) of these 20 cases was calculated as

$$\text{MAPE} = \frac{1}{20} \sum_{i}^{20} \mid \text{PE}_\text{i} \mid \tag{17}$$

and is shown in Table 1.

Out of all three approaches, the spline interpolation works best for rows Axial 1 and Axial 2, producing 4.91% error for Axial 1 and 2.93% for Axial 2 on average. The spline interpolation likely works better than the linear interpolation in particular because in this paper, pressures were interpolated. As discussed above, pressures and force are, however, not linearly correlated but slightly nonlinear. This nonlinearity is likely captured better by the spline interpolation. This is also visible by looking at the percentage error of the linear interpolation, which is almost always positive, because the linear interpolation underestimates the actual pressure due to the pressure's nonlinearity, therefore overestimating the life. Further, the deformation behavior of the surrounding structures influence the load distribution and cause non-linear behavior w.r.t the moment $M$, which can be captured better with the spline interpolation.

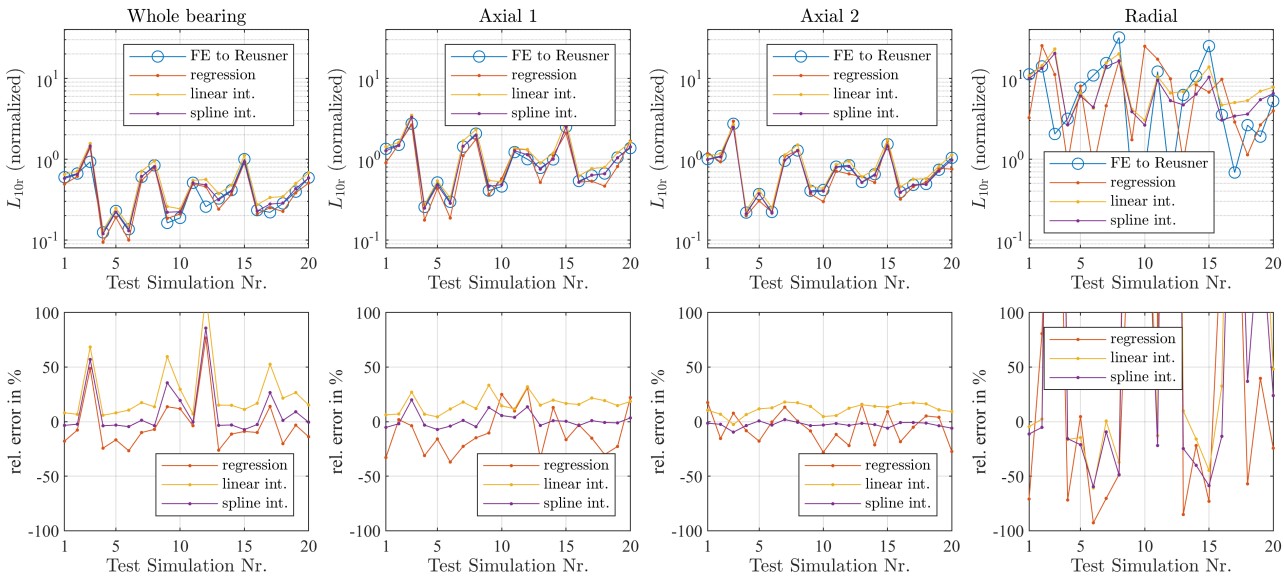

**Figure 20.** Top row: $L_{10\mathrm{r}}$ of all 20 test simulations, normalized, FE and interpolated. Bottom row: Percentage error for different approaches.

The interpolated pressures of Axial rows 1 and 2 are shown in Fig. 21 for test simulation 3. For this case, the errors for $L_{10\mathrm{r}}$ of Axial 1 are 3.2 % and those for Axial 2 are 3.6 %. The interpolated distribution can be seen to match the FE reference closely. Slight differences even out in the calculation of $L_{10\mathrm{r}}$.

For the radial row, accurate results are not achieved with any of the methods. Even using the spline interpolation, deviations of over 190 % exist on average for the interpolated life of the radial row. However, because the radial row makes up only a relatively small portion of the total bearing life, the spline approximation of the total bearing life achieves only 13.71 % deviation on average, even despite the poor performance with respect to radial row life interpolation.

Analyzing the 20 cases here showed that even small deviations in bending moment $M$ and load angle $\beta$ away from the grid points can result in very large qualitative and quantitative changes of the pressure (and force) distribution in the radial row. An example case is shown in Fig. 22, where a test simulation and two grid simulations are plotted. The test simulation is number 3 from Table B1, whose interpolation results in very poor results, see Fig. 20. The simulation has a load angle of $\beta = 62.5°$ and a bending moment of $M = 0.4376 \cdot M_{\max}$. Both of these values are very close to the also displayed grid simulations number 23 and 24, both of which have load angle $\beta = 60°$ and $M = 0.4 \cdot M_{\max}$. The $F_z$ value of test simulation 3 lies in between the extremes $-F_z$ and $+F_z$ from grid simulation 23 and 24, respectively. Despite this similarity, test simulation 3 can be seen to have higher pressures than either grid simulation 23 or 24. Aside from these quantitative differences, even the qualitative behavior is different, with grid simulation 24 carrying load at $0°$ and $90°$ where the other two simulations are carrying none, but carrying almost none at $180°$ where the other two are carrying a lot. Since the influence of $F_z$ was checked in Sect. 3.1.2, it is unlikely that nonlinear behavior of the pressure in between the extremes of $F_z$ explains this behavior.

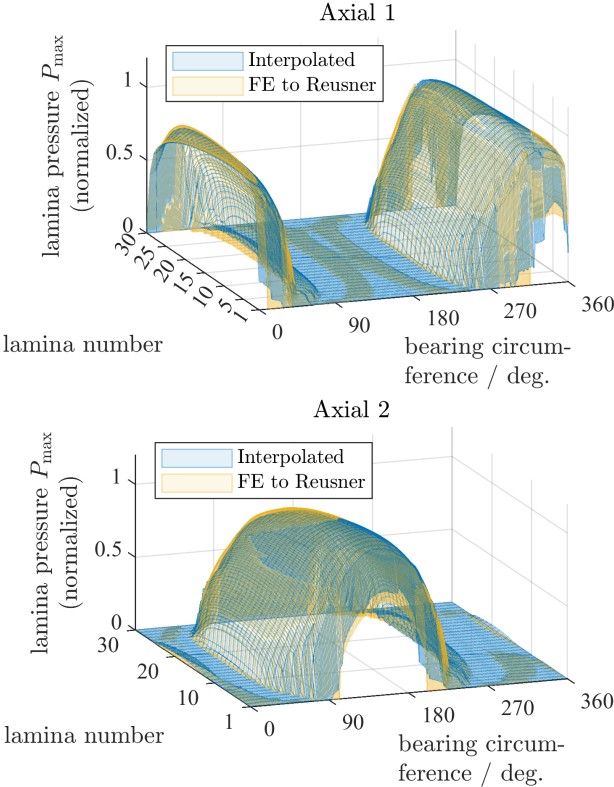

**Figure 21.** Test simulation 4: Interpolated pressure distribution vs. FE to inverted Reusner processed pressure distribution. Pressures normalized to maximum pressure of Reusner processed results of test simulation 4.

This indicates that the reason the radial row cannot be interpolated well may be due to insufficient available data for an accurate interpolation, that is to say, the radial row may be affected by more DOF than just those three used in this study. As discussed in Sect. 3, the grid was used based on a previous publication (Menck et al., 2020) that has been performed for a four-point bearing, which does not contain a radial row. The validity of $F_z$-based interpolation was explored in Sect. 3.1.2 and appeared to work well, but for more complex cases in which more than one DOF varies, the deformation may correlate worse with the three DOF chosen here. Ultimately, the pressure (and force) distribution that occurs in the radial row is very difficult to predict. To the current knowledge of the authors, it does not simply correlate with few DOFs but rather depends on the structural deformation of the entire bearing, which is affected by the interplay of all DOFs and the structural stiffness of the bearing.

The radial row therefore presents unfortunate properties:

- it makes up only a small portion of the total bearing life

- its calculation is very computationally expensive, because it contains a lot of rollers, and the inner and outer ring pressures are different and both have to be calculated

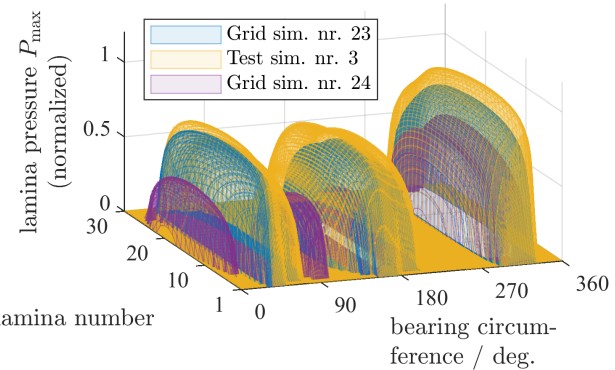

**Figure 22.** Radial pressures for grid simulation nr. 23 ($\beta = 60°$, $M = 0.4 \cdot M_{\mathrm{max}}$, $-F_z$), test simulation nr. 3 ($\beta = 62.5°$, $M = 0.4376 \cdot M_{\mathrm{max}}$, $+F_z \cdot 0.4294$), grid simulation nr. 24 ($\beta = 60°$, $M = 0.4 \cdot M_{\mathrm{max}}$, $+F_z$). These three cases highlight the difficulties in interpolating the radial row pressures, since test sim. 3 is close to grid simulations 23 and 24 w.r.t $M$ and $\beta$ and in between them w.r.t $F_z$, yet exceeds their pressures in many locations. Pressures normalized to maximum pressure of radial row in test simulation 3.

– its pressure (force) distribution is very difficult to predict and behaves in a very nonlinear fashion with respect to the DOF chosen in this study

    – the FE grid used in this paper may not be sufficient to describe it even though it seems very appropriate for the axial rows.

Due to the low influence on total bearing life, the results produced here are deemed acceptable, though further work on the
characterization of the radial row's life may be appropriate.

Based on the results in this section, the interpolation appears to work well in particular with the spline approach. The spline interpolation was thus used for the further calculations in the following sections.

## 5.2 Combined operating life

The lamina pressure interpolations that are explored above for 20 test cases are then performed for every single time step
$i = 1, 2, ..., I$ in the aeroelastic simulations. The life of the bearing is then calculated according to Sect. 4.1 for each of these time steps, using the interpolated pressures.

Now the individual time steps are combined into a total operating life of the turbine. This process takes into account the multipliers $x_i$ according to the Weibull distribution of the wind turbine class for DLC 1.2, and according to other, manufacturer-specific factors for the other DLCs. It also takes into account the movement that is performed in each time step, measured in
degrees.

The proportion of operating movement $\phi_i$ is determined via

$$\phi_i = \frac{s_i}{s_1 + s_2 + \ldots + s_I} \tag{18}$$

where $s_i = x_i \, |\, \theta_{i+1} - \theta_i \,|$, with $\theta_i$ being the pitch angle at time step $i$. The sum $s_1 + s_2 + \ldots + s_I$ then gives the amount of movement a pitch bearing performs during the desired life of the turbine, typically 20 to 30 years.

The variables $\phi_i$ are then used to weigh the lives $L_i$ of each time step $i$ to obtain the final combined life $L_{10}$

$$L_{10} = \frac{1}{\frac{\phi_1}{L_1} + \frac{\phi_2}{L_2} + \ldots + \frac{\phi_I}{L_I}} \tag{19}$$

Eq. 19 gives the life in millions of revolutions. This can be converted to life in years by multiplying

$$L_{10,\mathrm{y}} = \frac{1}{n_{\mathrm{rev/y}}} L_{10} \tag{20}$$

with $n_{\mathrm{rev/y}}$ giving the rotations performed per year, obtained by $n_{\mathrm{rev/y}} = \sum_i^I s_i / (360° \cdot T_{\mathrm{field}})$, where $T_{\mathrm{field}}$ is the planned turbine life, typically 20 to 30 years.

Further factors may be multiplied with this basic fatigue life yielding a modified fatigue life. These values may be based on experience of the manufacturer, as well as on properties of the bearing like its hardening depth, raceway hardness, or desired reliabilities other than 90 % used for $L_{10}$, see Stammler et al. (2024). The highest factor is a suggested value of 3, see Stammler et al. (2024).

## 5.3 Simplified life calculation

For double-row four-point bearings, the authors proposed in Menck et al. (2020) an adjustment of the simplified life calculation in the original NREL DG03 version of Harris et al.. This adjustment occurred by changing the variable $k_{\mathrm{M}}$ in the following equation

$$P_{\mathrm{a}} = 0.75 \cdot F_{\mathrm{r}} + F_{\mathrm{a}} + \frac{k_{\mathrm{M}} M}{D_{\mathrm{pw}}} \tag{21}$$

where $k_{\mathrm{M}}$ is modified based on the FE results of the particular bearing in question. The life is then calculated as $L_{10} = (C_{\mathrm{a}}/P_{\mathrm{a}})^3$ for ball bearings. The radial load is thus included in the dynamic axial load.

In the newer version of NREL DG03 (Stammler et al., 2024), the authors attempt to modify this equation for three-row roller bearings. They suggest to calculate the life of the radial row and that of the axial rows separately. For the radial row, they propose to use $L_{10,\mathrm{r}} = (C_{\mathrm{r}}/P_{\mathrm{r}})^{10/3}$, where $P_{\mathrm{r}} = F_{\mathrm{r}}$, and for the combined axial rows, they propose $L_{10,\mathrm{a}} = (C_{\mathrm{a}}/P_{\mathrm{a}})^{10/3}$ with $P_{\mathrm{a}} = F_{\mathrm{a}} + \frac{k_{\mathrm{M}} M}{D_{\mathrm{pw}}}$.

The results of this paper suggest that this is not necessary. The radial load $F_{\mathrm{r}}$ is very small compared to the load rating $C_{\mathrm{r}}$ of the radial row. Following the above described procedure, the life of the radial row is over 200 times as large as that of the axial row. This is incorrect: for the current example, the actual life of the radial row is only about 10-15 times as high as that of the axial rows, see Fig. 18. The bending moment and the resulting structural deformation of the bearing therefore appears to play a larger role in the loading of the radial row than the radial load $F_{\mathrm{r}}$ itself.

We therefore deviate from our recommendations in Stammler et al. (2024) and propose to keep a simple formula akin to Eq. 21 for three-row roller bearings due to the following reasons:

- The procedure described in Stammler et al. (2024) significantly overestimates the life of the radial row;

- Nonetheless, the influence of the radial row on total bearing life is much smaller than that of the axial rows;

- Even if the influence of the radial row were bigger, this would not be picked up by the procedure given in Stammler et al. (2024), because it would be due to structural deformation, not due to the radial load $F_r$;

- The procedure described in Stammler et al. (2024) is unnecessarily complicated for a simplified approach and yields no benefits to Eq. 21 due to the abovementioned reasons.

Thus, for an even more simplified approach than the NREL DG03, the life of the bearing $L_{10} = (C_a/P_a)^{10/3}$ can be determined from using just Eq. 4 for $C_a$ and Eq. 21 for $P_a$, if the factor $k_M$ is adjusted. This simplified approach can be useful for parametric studies, for example of the effect of different pitch bearings, wind speed distributions, or controllers.

It is possible for $k_M$ for a roller bearing to be bigger than for the double-row four-point bearing in Menck et al. (2020). One has to consider that a roller bearing has a considerably higher dynamic capacity $C_a$ than a double-row four-point (ball) bearing of comparable size. Due to this, the net effect of using a roller bearing may still yield an increase in life compared to the use of a double-row four-point bearing, even if $k_M$ is bigger.

## 6 Conclusions

The paper presented an approach to calculate the fatigue life of a three-row roller bearing that is used as a pitch bearing in a wind turbine. An FE model was presented and validated against experimental data from a real-scale pitch bearing test rig. The used FE modelling approach with 21 nonlinear spring elements per roller is a proper method for the simulation of realistic roller bearing deformation behaviour. Furthermore, this high number of spring elements allows an advanced implementation of the roller crowning and in turn leads to very similar pressure distributions compared to the Reusner model.

In order to narrow down the millions of aeroelastic time steps to a few FE simulations that can be calculated in reasonable time, a grid of FE simulations was determined for which the degrees of freedom of the bearing were reduced as much as possible. The three degrees of freedom that were retained are the resulting bending moment $M$, its load angle $\beta$, and the axial force $F_z$. For the axial force $F_z$, two extreme ends were simulated only for each grid case because analyses showed that the bearing behaved very linearly in between two extremes. The radial loads $F_x$ and $F_y$ where shown to correlate very closely with the bending moment components $M_y$ and $M_x$ and were therefore approximated with a linear fit. Blade 2 and 3 loads were averaged for each discrete load of blade 1.

This resulted in a grid of 72 FE simulations which envelop almost all load combinations of the entire aeroelastic data. For these 72 simulations, the contact pressures of all rollers were determined using a non-Hertzian contact algorithm based on Reusner (1977). Unlike the Hertzian rolling elements used in FE, the non-Hertzian algorithm allows for detection of edge stresses and is more accurate in general. A convergence analysis was carried out to determine the required amount of laminae for the non-Hertzian algorithm, which was determined to be 30 laminae, with an allowed error of 3 % compared to a reference simulation with 150 laminae. Since there were no severe cases of high edge stresses present in the bearing for all load cases

that have been simulated, the life calculated with the non-Hertzian post-processing was simular to that calculated with Hertzian pressures from the FE results.

Instead of interpolating (or approximating via a regression) the forces in between the grid points, pressures were interpolated (or approximated via a regression) instead, because the non-Hertzian algorithm would take too much time to compute each single time step. The interpolation approach was validated using 20 randomly sampled test cases around nominal turbine wind

speed. A cubic spline interpolation approach proved to be the best method to determine pressures in between the grid points, resulting in very low errors of 4.9 % and 2.9 % for the axial rows, and 193.6 % for the radial row, and a corresponding error of 13.7 % for the whole bearing life. The radial row life was very difficult to determine with all approaches, leading to the supposition that the grid may not be apt for the radial row because radial row pressures may mostly be affected by structural deformation. The radial row life therefore represents by far the biggest uncertainty in the final life, which was nonetheless

accepted for this calculation because its life is comparatively high even considering the uncertainty. Axial rows, which are at much higher risk of failure according to the calculation, were calculated with low uncertainty.

The present paper further discussed how to calculate the life of the bearing including all operating conditions. No final life was given due to confidentiality constraints. An approximation for the equivalent load of the adapted three-row roller bearing used in this paper was proposed. While the resulting equivalent load may be substantially higher than in a double-row four-

580 point bearing from a previous publication, this result must be interpreted considering the fact that roller bearings have a higher dynamic capacity to begin with for comparable dimensions, therefore resulting in a higher life nonetheless.

While the method described in this paper was carried out on one particular turbine, the authors assume that it would also be applicable for other wind turbines using three-row roller bearings as pitch bearings. Slight variations, e.g. in the amount of laminae or $k_{\mathrm{M}}$, may be necessary in this case.

*Data availability.* TEXT

Some of the exemplary plots use IWT7.5 data, which can be found under Popko. No further underlying data can be made available due to contractual constraints.

## Appendix A: Grid of FE simulations

The grid of FE simulations is given in Tables A1 and A2. Bending moments are given depending on the maximum bending
moment $M_{\mathrm{max}}$ that was simulated in this paper. Only the sign (positive or negative) is given for $F_z$. Note that the upper and
lower end of $F_z$ ($+F_z$ and $-F_z$, respectively) are different in absolute magnitude, i.e., the simulations were not symmetrical
around 0.

## Appendix B: Test simulations

The test simulations carried out are given here in a simplified manner. Actual test simulations were simulated including all 15
DOF. This includes 5 DOF of each rotor blade bearing of the turbine. These 15 DOF are summarized into 3 here for simpler
presentation, given in Table B1.

**Table A1.** Grid of FE simulations, part 1.

| Simulation Nr. | Load angle $\beta$ | Moment $M$ | Axial load $F_z$ |
|---|---|---|---|
| 1 | 0° | 20 % of $M_{\text{max}}$ | $-F_z$ |
| 2 | 0° | 20 % of $M_{\text{max}}$ | $+F_z$ |
| 3 | 0° | 40 % of $M_{\text{max}}$ | $-F_z$ |
| 4 | 0° | 40 % of $M_{\text{max}}$ | $+F_z$ |
| 5 | 0° | 60 % of $M_{\text{max}}$ | $-F_z$ |
| 6 | 0° | 60 % of $M_{\text{max}}$ | $+F_z$ |
| 7 | 0° | 80 % of $M_{\text{max}}$ | $-F_z$ |
| 8 | 0° | 80 % of $M_{\text{max}}$ | $+F_z$ |
| 9 | 0° | 100 % of $M_{\text{max}}$ | $-F_z$ |
| 10 | 0° | 100 % of $M_{\text{max}}$ | $+F_z$ |
| 11 | 30° | 20 % of $M_{\text{max}}$ | $-F_z$ |
| 12 | 30° | 20 % of $M_{\text{max}}$ | $+F_z$ |
| 13 | 30° | 40 % of $M_{\text{max}}$ | $-F_z$ |
| 14 | 30° | 40 % of $M_{\text{max}}$ | $+F_z$ |
| 15 | 30° | 60 % of $M_{\text{max}}$ | $-F_z$ |
| 16 | 30° | 60 % of $M_{\text{max}}$ | $+F_z$ |
| 17 | 30° | 80 % of $M_{\text{max}}$ | $-F_z$ |
| 18 | 30° | 80 % of $M_{\text{max}}$ | $+F_z$ |
| 19 | 30° | 100 % of $M_{\text{max}}$ | $-F_z$ |
| 20 | 30° | 100 % of $M_{\text{max}}$ | $+F_z$ |
| 21 | 60° | 20 % of $M_{\text{max}}$ | $-F_z$ |
| 22 | 60° | 20 % of $M_{\text{max}}$ | $+F_z$ |
| 23 | 60° | 40 % of $M_{\text{max}}$ | $-F_z$ |
| 24 | 60° | 40 % of $M_{\text{max}}$ | $+F_z$ |
| 25 | 60° | 60 % of $M_{\text{max}}$ | $-F_z$ |
| 26 | 60° | 60 % of $M_{\text{max}}$ | $+F_z$ |
| 27 | 60° | 80 % of $M_{\text{max}}$ | $-F_z$ |
| 28 | 60° | 80 % of $M_{\text{max}}$ | $+F_z$ |
| 29 | 60° | 100 % of $M_{\text{max}}$ | $-F_z$ |
| 30 | 60° | 100 % of $M_{\text{max}}$ | $+F_z$ |
| 31 | 90° | 0 % of $M_{\text{max}}$ | $-F_z$ |
| 32 | 90° | 0 % of $M_{\text{max}}$ | $+F_z$ |
| 33 | 90° | 20 % of $M_{\text{max}}$ | $-F_z$ |
| 34 | 90° | 20 % of $M_{\text{max}}$ | $+F_z$ |
| 35 | 90° | 40 % of $M_{\text{max}}$ | $-F_z$ |
| 36 | 90° | 40 % of $M_{\text{max}}$ | $+F_z$ |

**Table A2.** Grid of FE simulations, part 2.

| Simulation Nr. | Load angle $\beta$ | Moment $M$ | Axial load $F_z$ |
|---:|---|---|---|
| 37 | 90° | 60 % of $M_{\mathrm{max}}$ | $-F_z$ |
| 38 | 90° | 60 % of $M_{\mathrm{max}}$ | $+F_z$ |
| 39 | 90° | 80 % of $M_{\mathrm{max}}$ | $-F_z$ |
| 40 | 90° | 80 % of $M_{\mathrm{max}}$ | $+F_z$ |
| 41 | 90° | 100 % of $M_{\mathrm{max}}$ | $-F_z$ |
| 42 | 90° | 100 % of $M_{\mathrm{max}}$ | $+F_z$ |
| 43 | 120° | 20 % of $M_{\mathrm{max}}$ | $-F_z$ |
| 44 | 120° | 20 % of $M_{\mathrm{max}}$ | $+F_z$ |
| 45 | 120° | 40 % of $M_{\mathrm{max}}$ | $-F_z$ |
| 46 | 120° | 40 % of $M_{\mathrm{max}}$ | $+F_z$ |
| 47 | 120° | 60 % of $M_{\mathrm{max}}$ | $-F_z$ |
| 48 | 120° | 60 % of $M_{\mathrm{max}}$ | $+F_z$ |
| 49 | 120° | 80 % of $M_{\mathrm{max}}$ | $-F_z$ |
| 50 | 120° | 80 % of $M_{\mathrm{max}}$ | $+F_z$ |
| 51 | 120° | 100 % of $M_{\mathrm{max}}$ | $-F_z$ |
| 52 | 120° | 100 % of $M_{\mathrm{max}}$ | $+F_z$ |
| 53 | 150° | 20 % of $M_{\mathrm{max}}$ | $-F_z$ |
| 54 | 150° | 20 % of $M_{\mathrm{max}}$ | $+F_z$ |
| 55 | 150° | 40 % of $M_{\mathrm{max}}$ | $-F_z$ |
| 56 | 150° | 40 % of $M_{\mathrm{max}}$ | $+F_z$ |
| 57 | 150° | 60 % of $M_{\mathrm{max}}$ | $-F_z$ |
| 58 | 150° | 60 % of $M_{\mathrm{max}}$ | $+F_z$ |
| 59 | 150° | 80 % of $M_{\mathrm{max}}$ | $-F_z$ |
| 60 | 150° | 80 % of $M_{\mathrm{max}}$ | $+F_z$ |
| 61 | 150° | 100 % of $M_{\mathrm{max}}$ | $-F_z$ |
| 62 | 150° | 100 % of $M_{\mathrm{max}}$ | $+F_z$ |
| 63 | 180° | 20 % of $M_{\mathrm{max}}$ | $-F_z$ |
| 64 | 180° | 20 % of $M_{\mathrm{max}}$ | $+F_z$ |
| 65 | 180° | 40 % of $M_{\mathrm{max}}$ | $-F_z$ |
| 66 | 180° | 40 % of $M_{\mathrm{max}}$ | $+F_z$ |
| 67 | 180° | 60 % of $M_{\mathrm{max}}$ | $-F_z$ |
| 68 | 180° | 60 % of $M_{\mathrm{max}}$ | $+F_z$ |
| 69 | 180° | 80 % of $M_{\mathrm{max}}$ | $-F_z$ |
| 70 | 180° | 80 % of $M_{\mathrm{max}}$ | $+F_z$ |
| 71 | 180° | 100 % of $M_{\mathrm{max}}$ | $-F_z$ |
| 72 | 180° | 100 % of $M_{\mathrm{max}}$ | $+F_z$ |

**Table B1.** FE test simulations.

| Simulation Nr. | Load angle $\beta$ | Moment $M$ | Axial load $F_z$ |
|---:|---|---|---|
| 1 | 52.68° | 56.61 % of $M_{max}$ | 53.66 % of $+F_z$ |
| 2 | 55.49° | 55.29 % of $M_{max}$ | 62.19 % of $+F_z$ |
| 3 | 62.50° | 43.76 % of $M_{max}$ | 42.94 % of $+F_z$ |
| 4 | 64.70° | 89.37 % of $M_{max}$ | 41.87 % of $+F_z$ |
| 5 | 66.40° | 76.45 % of $M_{max}$ | 62.52 % of $+F_z$ |
| 6 | 69.09° | 88.16 % of $M_{max}$ | 66.34 % of $+F_z$ |
| 7 | 74.63° | 58.10 % of $M_{max}$ | 68.52 % of $+F_z$ |
| 8 | 75.01° | 53.47 % of $M_{max}$ | 69.52 % of $+F_z$ |
| 9 | 75.42° | 76.98 % of $M_{max}$ | 39.13 % of $+F_z$ |
| 10 | 93.64° | 77.04 % of $M_{max}$ | 35.54 % of $+F_z$ |
| 11 | 93.65° | 61.31 % of $M_{max}$ | 70.74 % of $+F_z$ |
| 12 | 100.71° | 61.73 % of $M_{max}$ | 45.85 % of $+F_z$ |
| 13 | 100.82° | 69.80 % of $M_{max}$ | 66.12 % of $+F_z$ |
| 14 | 101.19° | 65.07 % of $M_{max}$ | 72.68 % of $+F_z$ |
| 15 | 101.85° | 50.80 % of $M_{max}$ | 69.06 % of $+F_z$ |
| 16 | 104.19° | 76.23 % of $M_{max}$ | 57.73 % of $+F_z$ |
| 17 | 106.25° | 71.77 % of $M_{max}$ | 51.29 % of $+F_z$ |
| 18 | 107.37° | 71.10 % of $M_{max}$ | 54.42 % of $+F_z$ |
| 19 | 110.58° | 62.86 % of $M_{max}$ | 55.65 % of $+F_z$ |
| 20 | 110.81° | 57.47 % of $M_{max}$ | 46.81 % of $+F_z$ |

*Author contributions.* OM: Conceptualization, Methodology, Software, Validation, Investigation, Writing - Original Draft, Visualization; FS: Methodology, Software, Validation, Writing - Original Draft; MS: Writing - Review & Editing

*Competing interests.* The authors declare no conflicts of interest.

*Acknowledgements.* We gratefully acknowledge funding from the European Union's Horizon 2020 research and innovation programme under grant agreement No 791875, project name ReaLCoE. We thank GE Wind for allowing this publication and we thank Valentin Radigois for his useful feedback and support. We thank Martin Geibel for generating the plots of the FE validation.

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
