# Peer review of "Rolling contact fatigue calculation of a three-row roller pitch bearing in a wind turbine"

_Wind Energy Science, 2025_

## Community Comment (CC1)

In this paper, the authors describe a methodology for a rolling contact rolling contact fatigue life calculation for a three-row roller bearing. An example calculation is provided. Several interesting clarifications compared to the approach described in the NREL DG03 are made, along with some interesting technical results gleaned from the example calculations. I was going to read this paper in depth no matter what, and in doing so, figured I would make some hopefully helpful clarifying comments.

Abstract

- Line 8: I'm not sure I understand "the bearing is modified slightly" pertains to the model or the actual bearing (or both). I believe this really relates to the aspect of confidentiality in the next sentence. To be honest, I think stating "Exemplary calculations…" states all that is needed here, as the actual dimensions or other structural and material aspects of the bearing are not given in the paper. Another option might be stating "Exemplary calculations are carried out using a slightly modified version of an extensively validated FE model of a three-row roller bearing of a wind turbine."
- Lines 9-10: I understand this sentence "Since…confidentiality…", but I'll admit it feels a bit odd to state in the Abstract. I recommend it be deleted here and only stated in the text if really necessary. Indeed, this is done on line 56 and it feels sufficient and appropriate there.
- In the Abstract, I do recommend adding a statement better summarizing the main findings in the paper. From the Conclusions my main takeaway was that "The axial rows were found to have a much lower fatigue life than the radial row, and thus the axial rows are the main determinant of the fatigue life of the bearing. They also were shown to have a much lower uncertainty than the radial row."

1 Introduction

- Lines 21-22: Strictly speaking, cyclic changes in subsurface stress don't *cause* inclusions or material defects themselves to grow, so I think slightly modifying this sentence to something like "Cyclic changes in sub surface stress near inclusions or material defects cause microcracks that grow into larger, macroscopic spalls."

2.1 FE bearing model and validation

- Lines 106-108: I'm not sure I understand the meaning of "The scatter bars in the plots indicate the fluctuation in the measured signals which is caused by the pitch movements of the bearing". As is, I took this to mean that the uncertainty would be in the horizontal axis rather than the vertical. Rather, I interpret this statement to mean that the rolling elements may not be in exactly the same location relative to the strain gauge as the bearing rotates (pitches) and the rollers orbit – I would assume this to possibly be related to some amount of sliding, even very small. Is that correct? So maybe "…caused by small differences in the locations of the rollers relative to the strain gauge as the bearing pitches" is a clearer way to say it? I can understand then that for smaller bearings this effect is more pronounced (as the size of the gauge is a larger proportion to the bearing circumference). I also think adding "…differently pronounced between both test bearings in Figure 6" would be helpful in line 108.

3.1.1 Approximation of Fx and Fy for blade 1

- I think Section 3.1.1 is quite interesting as it's the first time I have seen anyone look at and propose this. However, I wonder, if one is doing the aeroelastic simulations anyway, what is the value of approximating Fx and Fy? Can the authors comment? Is this maybe because if one is interested in relating field testing measurements of blade root loads to RCF calculations, it is only really possible to measure the blade root moment and not the force? If this is the intent, it might be worthwhile mentioning it.

**3.1.2 Choice of grid simulation points for Fz for blade 1**

- Line 212: A very minor point that the statement "Since high forces are those that affect the life the most" is really only partially true – it's really the combination of force and the number of cycles at that force that affect life the most. At least the rolling contact fatigue life. Or maybe the authors intend this statement to be generally the "life" due to all failure modes, rather than just the rolling contact fatigue life.

**3.3 Loads at blades 2 and 3**

- Line 240: Can a citation for "Note this influence of blade 2 and 3 is more significant for other damage modes such as ring cracks" be added?

**4 Rolling contact fatigue life for individual load cases**

- Line 256: The NREL DG03 describes more than 1 method to calculate rolling contact fatigue life, so I recommend this sentence be revised to "The rolling contact fatigue life calculation adheres closely to the ISO/TS 16281-based methodology described in the NREL DG03 (Stammler et al. 2024)."
- Line 263: I'm not sure if the adjective "operational" for fatigue life is really needed, compared to the other uses in the document of simply "fatigue life" or "rolling contact fatigue life". If "operational" does not have an intended meaning, I recommend deleting it.

**5.2 Combined operating life**

- Line 493: What is the variable $x\_i$?
- Line 501: I am often struck of our practice of doing quite detailed load and pressure calculations, but then when calculating a modified fatigue life, we use relatively large modification factors that are "based on experience". In the case of pitch bearings, the NREL DG03 suggests $a\_srv = 3$. Having said that, outside of a relatively small community I'm not sure how many end users are aware of such estimates when compared to the basic fatigue life calculations themselves. With that opinion in mind (and a bit of humility and honesty), I recommend revising this sentence to at lease allude to this with something like "Further factors may be multiplied with this basic fatigue life yielding a modified fatigue life. The highest is a suggested value of 3 based on experience of the manufacturer, as well as on……..(Stammler et al. 2024)."

**5.3 Simplified life calculation**

- I suggest adding a clarification to the middle of the paragraph from lines 510 to 513 summarizing the NREL DG03 simplified calculation method, based on previous correspondence with the authors. The main reason being that Eq. 13 refers to m = 1 to 3, with the axial rows (m = 1 and 2)

and radial row (m = 3). So, a reader is again expecting m = 1 to 3 here. However, my understanding of the authors intent is that only two parts are used: an axial row (m =1) and the radial row (m=3). That is, the difference is that only 1 of the axial rows are used, not that the radial and axial rows are calculated separately (they are calculated separately both in section 4.1 and here in section 5.3). Thus, using consistent terminology as section 4.1 and Eq. 13 but highlighting the important difference, I believe it is clearer to state here "They propose calculating the life of only one of the two axial rows $L10,2 = (Ca/Pa)^{10/3}$ because it is assumed only one axial row at a time carries the axial load, where Pa = ..., and combining it with the life of the radial row $L10,3 = (Cr/Pr)^{10/3}$, where Pr = ... The life of the entire bearing is then calculated using Eq. 13 from just these two results (i.e. m = 2 to 3)." I believe this is the clearest description.

- Line 516: Another minor clarification, difference in radial load life between the methods is valid for the bearing and turbine loads studied in this paper, but I agree it is a trend likely to occur in other cases. With that, I recommend adding "This is incorrect: for the current example, the actual life of the radial row...."

- Line 527: Similar to my previous comment regarding lines 510 to 513, I recommend another clarification here such as "Thus, for an even more simplified approach than the NREL DG03, the life of the bearing $L10 = (Ca/Pa)^{10/3}$ can be determined using from just Eq. 4 for Ca and Eq.21 for Pa, if the factor kM is adjusted. This simplified approach can be useful for parametric studies, for example of the effect of different pitch bearings, wind speed distributions, or controllers."

6 Equivalent time for an accelerated fatigue life test on a pitch bearing test rig

- In general, this section "sticks out" a bit, as it's not really connected to the rest of the paper. I wonder if it would be worthwhile to highlight the fact that such a fatigue life test could be useful in assessing uncertainties with the radial row fatigue life, or the factor kM. I understand from the rest of the manuscript that the axial rows have a lower fatigue life....I suppose...but can one really say that confidently with such a high degree of uncertainty on the radial row fatigue life? Can something be added here that better connects this section with the rest of the paper?

7 Conclusions

- Line 549: Please add "...approach to calculate the fatigue life of..."

Minor grammatical comments:

- Line 32 (and elsewhere): ISO/TS 16281 (2008 edition) was recently revised and upgraded from a technical specification to international standard as ISO 16281 (2025 edition). It may be worthwhile considering whether to refer to the well-known TS versus the IS, depending on whether or not there are any relevant differences for this manuscript.
- Lines 39 and 45 (and elsewhere): "Keller, Jonathan and Guo, Yi" should read "Keller and Guo, 2022"
- Line 47 (and elsewhere): "Stammler et al." should be "Stammler et al., 2024".

---

## Author Comment (AC1)

Dear reviewer,

thank you for reading the paper and thank you for your feedback. You can find responses to your comments and questions below in blue.

Comments

Overall remarks

It is a well-written paper with enough material to form a substantial contribution to the pitch bearing literature, particularly as there is a dearth of papers about three row roller bearings. As it is a methodology paper, it should be relatively easy to follow the procedure adopted which it is. The flow of the methodology has been clearly illuminated by the well-separated chapters. The reasoning for the methodology has been illuminated properly with enough validation done for each aspect of the method. There are some minor issues which need to be corrected highlighted in the next section. Since the authors validated this method with its relevant assumptions for a specific wind turbine, it may also be good to describe or postulate if this method will work in general for wind turbines with three row roller bearings as well (as much as possible within the limits of confidentiality).

Thank you for the kind words. We believe the method shown is also applicable for other wind turbines that use three-row roller bearings as pitch bearings. Most of the assumptions we took should also hold true for other wind turbines. This information was added to the conclusions.

Specific comments

1. Line 7: The modification of the bearing is mentioned but never discussed further (except in the conclusion). Please either include a line or so in the main text about this. Even if it is confidential, possibly mention this.

   The modification that was undertaken is confidential, this information was added to the introduction.

2. Line 81: The choice of 21 slices for the FEM model is not explained. Why was this value chosen? Is it standard within the extension?

   It was the maximum possible value of springs in the extension at the time we wrote the paper, this information was added to the manuscript. (Side note: Further analyses undertaken by us later show that the amount of laminae used in FE has little impact on the results of the Reusner submodel; this will be published in a separate paper in the future)

3. Line 83: "contacting roller and raceway" may be a better formulation.

   Changed as suggested.

4. Line 88: Unless confidential, possibly mention if the split happens in the inner or outer ring and approximate location. I see a split in Figure 1; this could be highlighted there, too.

The location of the split is unfortunately confidential. Figure 1 may or may not be representative of the bearing we calculated. This information was added and the split in Figure 1 is now referred to for readers unfamiliar with the concept.

5. Line 198: The reference "Stammler et al.": Please date these references as there are multiple from the same author.

The date was added (it is from 2024, specifically this sentence is citing the new NREL DG03)

6. Line 316-318: It is mentioned that the Reusner algorithm gives a more accurate load distribution than the FE simulation. It is mentioned later on about it being due to the Hertzian nature of contact algorithm ("For these 72 simulations, the contact pressures of all rollers were determined using a non-Hertzian contact algorithm based on Reusner (1977). Unlike the Hertzian rolling elements used in FE, the non-Hertzian algorithm allows for detection of edge stresses and is more accurate in general."). To make it clearer, the hertzian nature of the FEM model can be explained before this part of the text.

The sentences "Pressures of these laminae can be calculated using Hertzian theory. However, Hertzian theory simplifies the roller-race contact and may underestimate the real pressure" were added prior to the cited sentences. Furthermore, the wording "This returns a more accurate load distribution" was changed to "This returns a more accurate pressure distribution" to clarify that pressure is the variable of interest here. (This is because, strictly speaking, the load-deflection relationship used in FEM based on ISO 16281 is based on Palmgren, and only the corresponding pressure calculation is based on so-called Hertzian theory).

7. Line 517: About the bending moment and resulting structural deformation playing a larger role in the loading of the radial row than the radial load itself: How exactly does structural deformation happen in this case? What is a possible reason for this? Can there be numbers put to this (if possible)?

Structural deformation causes the rings of the bearing to deform in ways that are unfortunately difficult to predict without detailed FE simulations (refer to Fig. 4, 5, and 6). In this case, the two contacting points of the radial rollers (the raceway on the "nose" of one ring and the raceway surface on the other) deflect, too, as a result of the global load, in particular the bending moment as it is the highest load acting on the bearing. This deflection of the "nose" and the other contact point of the radial row (evidently) affects the strange, unpredictable load distribution on the radial row more so than the radial force.

Putting numbers to this is unfortunately very difficult. In this paper, we are attempting that with these sentences:

*The radial load Fr is very small compared to the load rating Cr of the radial row. Following the above described procedure, the life of the radial row is over 200 times as large as that of the axial row. This is incorrect: the actual life of the radial row is only about 10-15 times as high as that of the axial rows, see Fig. 18.*

We are currently not aware of a better way to put numbers to this. The above relative difference may very well differ for other bearings on other turbines, too.

8. Line 527: By a simplified approach like in Eq. 21, do you mean to say that only the axial load (ignoring the Fr mentioned in Eq.21) is to be considered here? Also, it could also be clarified then that since it is a roller bearing, the 10/3 exponent is to be used. Also, for the sake of clarity, is Ca of a single axial row to be used for life?

No, the load Fr mentioned in Eq. 21 should also be included. We call this approach simplified because it is much simpler than the above described procedure that uses FE simulations, Reusner submodeling, etc.

We clarified that 10/3 should be used, thank you for the note.

For a three-row roller bearing, Ca of the entire bearing is always the Ca of a single axial row. This is because the bearing is a so-called double-direction bearing, where an *axial* load only loads one of the rows at a time but never both at once. For bending moments, both rows are loaded at once; but as the definition of Ca refers to an axial load, only one row's Ca is used as the Ca for the entire bearing. This definition follows ISO 281: Refer to ISO 281:2007, Sec. 8.1.1: *The basic dynamic axial load rating for single-row, single-direction or double-direction thrust roller bearing is given by Ca = [...]*

---

## Author Comment (AC2)

Dear reviewer,

thank you for reading the paper and thank you for your feedback. You can find responses to your comments and questions below in blue.

General remarks:

Well explained and clear structured paper.

Thank you for the kind words.

The graphics are too small and difficult to read.

Most of the graphics use a similar font size to the manuscript text, for images that are difficult to read, the graphics are all high-quality images and zooming in is possible. As there is a lot of information contained in the images we had to make do with the available space, unfortunately.

Some sources only given with name, no further information (year etc.) à e.g. "see Stammler et. al."

Corrected where we found this issue (also for Keller, Jonathan and Guo, Yi)

Will the use of a full 3D model of the bearing and the time series allow to predict the circumferential location of the rolling contact fatigue failure?

While this was not the focus of this paper, other works by the authors (see https://doi.org/10.1115/1.4055916) can be used to find the area of highest damage likelihood. Of course, with fatigue being a statistical phenomenon, an exact location can never be foreseen.

Data Quality "Fair" due to the confidentiality requirement.

In detail:

Line 7: In comparison to ball bearings, the calculation of roller bearings [..] leading to significantly more degrees of freedom.

A Ball bearings also have many degrees of freedom, if modelled with balls having realistic kinematic behaviour (instead of springs and fixed contact angle) since the contact angle ball to raceway can differ from ball to ball and is strongly load dependent.

This is true for FE modeling, but this sentence was specifically meant to refer to the DOFs in the fatigue calculation. Changed the "[...] the calculation of roller bearings [...]" to "[...] the rolling contact fatigue calculation of roller bearings [...]" to highlight this point.

Line 8: For this paper, the bearing is modified slightly: What does modified slightly mean? Please describe more detailed, if possible.

Unfortunately the details of the modification are confidential. The information that the details are confidential was added to the manuscript.

Line 56-58: Does the roller profile used for the calculation differ between the "confidential" turbine pitch bearing and the more "generale" bearing? If so, how is the influence onto the "normalized" results?

This information is unfortunately confidential.

Line 65: But is it possible to get the ISO life calculations **quantitively** reliable?

In the scope of this paper that's a question we won't answer (yet), but generally we believe so. Look out for future publications from us on this topic.

Line 81-83: Can you explain why 21 springs are used in the FE model, whereas ISO/TS16281 requires min. 30 slices in their roller slice model?

At the time of writing the paper the extension only allowed a maximum of 21 slices, which is why this value was chosen. We included this information in the manuscript. To ensure that this does not falsify the results, we undertook analyses (including custom-built FE models with 31 slices) which showed that the amount of slices in the FE model has little influence on the Reusner submodel so long as the Reusner submodel has sufficient slices; this will be published in a separate paper in the future.

Note also that the Reusner submodel, the results of which are used for the life calculation, did include 30 slices.

Line 88: [..] This split of the ring is considered in the model and the surfaces are connected to each other by an internal frictional contact. [..] Axial clamping force? Bolt preload? Influence bolt preload on roller preload?

The axial clamping force is driven by the bolt preload of the bolts. Each bolt along the circumference is preloaded with the same load which corresponds to the target mounting bolt preload. The higher the bolt preload the higher the preload of the axial rollers on both rows of the bearing.

Line 166-167: Influence of pitch angle might only become more important if the blade model differs circumferentially from its mechanical properties. How is the blade root stiffness modelled, uniform or non-uniform, anisotropic?

The orthotropic material behavior of the blade root / blade dummy composite material is considered in the model. The material model contains direction dependent Young's moduli, Poisson's ratios and shear moduli. For the blade root a blade dummy is used as written in a manuscript, however, further details on this subject are confidential.

Line 220: again the question about blade root model and physical properties around circumference - uniform or non-uniform

As explained above, for the blade root a blade dummy was used. Further details on this subject can unfortunately not be shared.

Line 296-298: Is there a difference for small back and forward motion compared to full rotations in calculating dynamic equivalent load and bearing lifetime? If so, further explanation would help. Have time steps with negligible rotation been ignored, if so what is the minimum pitch angle to be considered?

Time steps with negligible rotation have not been ignored, as there is no clear limit for what constitutes a "negligible" rotation known to us.

Yes, generally, there is a difference for small back and forward motion compared to full rotations in calculating dynamic equivalent load and bearing lifetime. When a bearing is fully rotating, the rotating ring is continuously rotating through the highest loaded zone. All spots on the circumference of the rotating ring thus go through the load zone. When it is oscillating back and forth, only one area on the circumference will be in the highest loaded zone. This is discussed in detail in Menck and Stammler (2024) and we added the following sentence to the manuscript to point this out:

*These small oscillations cause the oscillating ring to be almost stationary w.r.t. the load, differing from a rotating bearing, where the rotating ring rotates relative to the load.*

Line 310: why 21 laminae in FE compared to min. 30 as per ISO 16281

As explained above, this is in part due to limitations in the extension at the time of writing this paper, however, we ensured that it does not affect the results, details on which will be published in the future. The Reusner submodel does fulfill the minimum requirement of 30 slices.

Figure 14: Graphics of convergence analysis in not convincing compared to line 349. The graphics do not show why 30 lamina is a suitable choice.

They show that for 30 laminae, L10r is within a +-3% range of the reference result using 150 slices. This was the intention of the figure.

Line 405: What is the expected difference in rating life between analysis done on time steps compared to load bins?

This is impossible to answer generally as it highly depends on the binning process. In WES - Review of rolling contact fatigue life calculation for oscillating bearings and application-dependent recommendations for use, Figure 8, we are doing a comparison, but we would assume that for other choices of bins the result can be quite different.

Line 516: What impact will the radial row, at 10-15 times the life of the axial raceways, have on the combined rating life?

A very small impact. As you can see in Fig. 20, Column "Radial", even though the radial row approximations are sometimes wildly off the reference ("FE to Reusner"), the combined life in the "Whole bearing" column deviates very little.

Line 519-520: Will radial load be added to eq. ax. load Pa and compared with axial load capacity Ca? Further explanation required for formula.

Yes, that's the way Eq. 21 is generally employed. Added the sentence "The radial load is thus included in the dynamic axial load" after Eq. 21 to clarify this point.

Line 529: Further explanation required, also in relation to formula above. Normally ball bearings have higher k factor on moments compared to axial rows of 3 row roller bearings if calculating axial and radial raceways separately on 3 row roller bearings.

We are unsure which k factor you are referring to here, as this calculation is the first we are aware of for a three-row roller bearing in the published literature.

Line 566: What would be a high edge stress?

A significant increase of the stresses at the roller edges as compared to the roller center. We are unaware of any strict definitions (e.g., 50% higher than the roller center… or similar), but the phenomenon of high edge stresses is generally recognized in the literature, see ISO 16281, Harris and Kotzalas (2007), etc.

Line 581: What is the expected failure mode of a three-row roller bearing

We believe this question is beyond the scope of this paper, but rolling contact fatigue is one possible failure mechanism among a number of possible failure mechanisms.

---

## Author Comment (AC3)

Dear reviewer,

thank you for reading the paper and thank you for your feedback. You can find responses to your comments and questions below in blue.

In this paper, the authors describe a methodology for a rolling contact rolling contact fatigue life calculation for a three-row roller bearing. An example calculation is provided. Several interesting clarifications compared to the approach described in the NREL DG03 are made, along with some interesting technical results gleaned from the example calculations . I was going to read this paper in depth no matter what, and in doing so, figured I would make some hopefully helpful clarifying comments.

Thank you for the kind words and for your efforts.

Abstract

- Line 8: I'm not sure I understand "the bearing is modified slightly" pertains to the model or the actual bearing (or both). I believe this really relates to the aspect of confidentiality in the next sentence. To be honest, I think stating "Exemplary calculations…" states all that is needed here, as the actual dimensions or other structural and material aspects of the bearing are not given in the paper. Another option might be stating "Exemplary calculations are carried out using a slightly modified version of an extensively validated FE model of a three-row roller bearing of a wind turbine."
  The modification pertains to the bearing model used for this paper only.
  The sentence was changed as suggested.

- Lines 9-10: I understand this sentence "Since…confidentiality…", but I'll admit it feels a bit odd to state in the Abstract. I recommend it be deleted here and only stated in the text if really necessary. Indeed, this is done on line 56 and it feels sufficient and appropriate there.
  Removed the note about confidentiality from the abstract as suggested.

- In the Abstract, I do recommend adding a statement better summarizing the main findings in the paper. From the Conclusions my main takeaway was that "The axial rows were found to have a much lower fatigue life than the radial row, and thus the axial rows are the main determinant of the fatigue life of the bearing. They also were shown to have a much lower uncertainty than the radial row."
  Added the sentence as suggested.

1 Introduction

- Lines 21-22: Strictly speaking, cyclic changes in subsurface stress don't *cause* inclusions or material defects themselves to grow, so I think slightly modifying this sentence to something like

  "Cyclic changes in sub surface stress near inclusions or material defects cause microcracks that grow into larger, macroscopic spalls."

  Changed as suggested

2.1 FE bearing model and validation

- Lines 106-108: I'm not sure I understand the meaning of "The scatter bars in the plots indicate the fluctuation in the measured signals which is caused by the pitch movements of the bearing". As is, I took this to mean that the uncertainty would be in the horizontal axis rather than the vertical. Rather, I interpret this statement to mean that the rolling elements may not be in exactly the same location relative to the strain gauge as the bearing rotates (pitches) and the rollers orbit – I would assume this to possibly be related to some amount of sliding, even very small. Is that correct? So maybe "…caused by small differences in the locations of the rollers relative to the strain gauge as the bearing pitches" is a clearer way to say it? I can understand then that for smaller bearings this effect is more pronounced (as the size of the gauge is a larger proportion to the bearing circumference). I also think adding "…differently pronounced between both test bearings in Figure 6" would be helpful in line 108.
  The strain gauges are in one constant position but as the bearing rotates, they fluctuate somewhat. We have observed this behavior on all our test rigs and it is not specific to the one in this paper, neither can we say for sure that it is due to the rolling elements' movement, as previous publications have shown that manufacturing anomalies within the tolerances can also affect the strain gauge results (https://doi.org/10.1016/j.finel.2024.104268). Differences in the scatter bars are observable in Figs 4-6, not only 6.

3.1.1 Approximation of Fx and Fy for blade 1

- I think Section 3.1.1 is quite interesting as it's the first time I have seen anyone look at and propose this. However, I wonder, if one is doing the aeroelastic simulations anyway, what is the value of approximating Fx and Fy? Can the authors comment? Is this maybe because if one is interested in relating field testing measurements of blade root loads to RCF calculations, it is only really possible to measure the blade root moment and not the force? If this is the intent, it might be worthwhile mentioning it.
  The intent here is just to be able to reduce the number of simulations that are being done in FE. It is only feasible to do a small number of FE simulations and therefore we need to reduce the DOFs as much as possible. This approximation of Fx and Fy is thus used to perform FE simulations that only use Mres, the load angle, and Fz as DOFs, but still include a "realistic" Fx and Fy by determining them based on the bending moment Mres and its load angle.

3.1.2 Choice of grid simulation points for Fz for blade 1

- Line 212: A very minor point that the statement "Since high forces are those that affect the life the most" is really only partially true – it's really the combination of force and the

number of cycles at that force that affect life the most. At least the rolling contact fatigue life. Or maybe the authors intend this statement to be generally the "life" due to all failure modes, rather than just the rolling contact fatigue life.

You are entirely correct in stating that high forces in combination with movement affect the life, but here we are discussing one given load distribution and we are commenting on the loads within that load distribution. In that context, it is not possible for the area around 180° to have significantly less movement than that around 90° unless (very) strange slippage effects occur.

Changed the sentence to "*Since highly loaded rolling elements are those that affect the life the most, this result is adequate for the life calculation*" to clarify this.

**3.3 Loads at blades 2 and 3**

- Line 240: Can a citation for "Note this influence of blade 2 and 3 is more significant for other damage modes such as ring cracks" be added?

  There is no clear source for this but " Multi-MW Blade Bearing Applications –Advanced Blade Bearing Design Process and Pitch Bearing Modul Development Trends"  by Daniel Becker, presented at IQPC 2023, states

  *Edgewise loads do not play a significant role for raceway fatigue, but do for ring / bolt fatigue …*

  Edgewise loads are influenced by blades 2 and 3, hence their effect on ring cracks. However as no clearer source could be provided we changed the sentence to

  *Note this influence of blade 2 and 3 is likely more significant for other damage modes such as ring cracks, as they influence the edgewise loads, which are known to affect ring cracks (Becker, 2023)*

**4 Rolling contact fatigue life for individual load cases**

- Line 256: The NREL DG03 describes more than 1 method to calculate rolling contact fatigue life, so I recommend this sentence be revised to "The rolling contact fatigue life calculation adheres closely to the ISO/TS 16281-based methodology described in the NREL DG03 (Stammler et al. 2024)."

  Changed as suggested

- Line 263: I'm not sure if the adjective "operational" for fatigue life is really needed, compared to the other uses in the document of simply "fatigue life" or "rolling contact fatigue life". If "operational" does not have an intended meaning, I recommend deleting it.
  Deleted as suggested.

**5.2 Combined operating life**

- Line 493: What is the variable $x_i$?
  It's a multiplier (based on the Weibull distribution), this is stated 5 lines before (*This process takes into account the multipliers $x_i$. […]*)

- Line 501: I am often struck of our practice of doing quite detailed load and pressure calculations, but then when calculating a modified fatigue life, we use relatively large modification factors that are "based on experience". In the case of pitch bearings, the NREL DG03 suggests a_srv = 3. Having said that, outside of a relatively small community I'm not sure how many end users are aware of such estimates when compared to the basic fatigue life calculations themselves. With that opinion in mind (and a bit of humility and honesty), I recommend revising this sentence to at lease allude to this with something like "Further factors may be multiplied with this basic fatigue life yielding a modified fatigue life. The highest is a suggested value of 3 based on experience of the manufacturer, as well as on……..(Stammler et al. 2024)."
  Changed to:
  *Further factors may be multiplied with this basic fatigue life yielding a modified fatigue life. These values may be based on experience of the manufacturer, as well as on properties of the bearing like its hardening depth, raceway hardness, or desired reliabilities other than 90 % used for L10, see Stammler et al. (2024). The highest factor is a suggested value of 3, see Stammler et al. (2024)*

5.3 Simplified life calculation

- I suggest adding a clarification to the middle of the paragraph from lines 510 to 513 summarizing the NREL DG03 simplified calculation method, based on previous correspondence with the authors. The main reason being that Eq. 13 refers to m = 1 to 3, with the axial rows (m = 1 and 2) and radial row (m = 3). So, a reader is again expecting m = 1 to 3 here. However, my understanding of the authors intent is that only two parts are used: an axial row (m =1) and the radial row (m=3). That is, the difference is that only 1 of the axial rows are used, not that the radial and axial rows are calculated separately (they are calculated separately both in section 4.1 and here in section 5.3). Thus, using consistent terminology as section 4.1 and Eq. 13 but highlighting the important difference, I believe it is clearer to state here "They propose calculating the life of only one of the two axial rows L10,2 = (Ca/Pa)^10/3 because it is assumed only one axial row at a time carries the axial load, where Pa = …, and combining it with the life of the radial row L10,3 = (Cr/Pr)^10/3, where Pr = … The life of the entire bearing is then calculated using Eq. 13 from just these two results (i.e. m = 2 to 3)." I believe this is the clearest description.

  No, there appears to be a misunderstanding, both axial rows are used for the calculation of L10,a. The load rating Ca is defined for a single row, since an axial bearing under an *axial* load would only carry on one of the rows. This definition follows ISO 281 (see our reply to RC1, comment 8). For a *bending moment*, both rows carry load, and hence for the *life*, the loads of both rows are considered.

- Line 516: Another minor clarification, difference in radial load life between the methods is valid for the bearing and turbine loads studied in this paper, but I agree it is a trend likely to occur in other cases. With that, I recommend adding "This is incorrect: for the current example, the actual life of the radial row…."
  Changed as suggested

- Line 527: Similar to my previous comment regarding lines 510 to 513, I recommend another clarification here such as "Thus, for an even more simplified approach than the NREL DG03, the life of the bearing L10 = (Ca/Pa)^10/3 can be determined using from just Eq. 4 for Ca and Eq.21 for Pa, if the factor kM is adjusted. This simplified approach can be useful for parametric studies, for example of the effect of different pitch bearings, wind speed distributions, or controllers."

Changed as suggested

6 Equivalent time for an accelerated fatigue life test on a pitch bearing test rig

- In general, this section "sticks out" a bit, as it's not really connected to the rest of the paper. I wonder if it would be worthwhile to highlight the fact that such a fatigue life test could be useful in assessing uncertainties with the radial row fatigue life, or the factor kM. I understand from the rest of the manuscript that the axial rows have a lower fatigue life….I suppose…but can one really say that confidently with such a high degree of uncertainty on the radial row fatigue life? Can something be added here that better connects this section with the rest of the paper?
We agree that the section is poorly connected to the rest of the paper and adds little to the available literature. Section 6 and all references to it were therefore removed from the manuscript.

7 Conclusions

- Line 549: Please add "…approach to calculate the fatigue life

of…"

Added as suggested

Minor grammatical comments:

- Line 32 (and elsewhere): ISO/TS 16281 (2008 edition) was recently revised and upgraded from a technical specification to international standard as ISO 16281 (2025 edition). It may be worthwhile considering whether to refer to the well-known TS versus the IS, depending on whether or not there are any relevant differences for this manuscript.
There are no relevant differences between ISO/TS 16281 and ISO 16281:2025 for this manuscript, therefore, all references were changed to ISO 16281. This does not include any references to the approach in NREL DG03 based on ISO/TS 16281, where the reference to ISO/TS 16281 remains, as here for example in the introduction:
*The calculation approach shown herein follows closely the abovementioned NREL DG03, the rolling contact fatigue calculation approach of which is based closely on ISO/TS 16281 (now replaced by ISO 16281).*

Neither this manuscript nor NREL DG03 are affected by any differences between ISO/TS 16281 and ISO 16281:2025 to the best knowledge of the authors.

- Lines 39 and 45 (and elsewhere): "Keller, Jonathan and Guo, Yi" should read "Keller and Guo, 2022"
corrected
- Line 47 (and elsewhere): "Stammler et al." should be "Stammler et al., 2024".
corrected